# Ratio-Variance Regularized Policy Optimization

Yu Luo[1]  Shuo Han[1]  Yihan Hu[1]  Lei Lv[2]  Huaping Liu[3]  Fuchun Sun[3]  Jianye Hao[4]  Dong Li[1]

## Abstract

Standard on-policy reinforcement learning relies on heuristic clipping to enforce trust regions, but this mechanism imposes a severe cost by indiscriminately truncating high-return yet high-divergence updates. We demonstrate that explicitly constraining the *policy ratio **variance*** provides a principled local approximation to trust-region constraints, eliminating the need for binary hard clipping. By acting as a distributional "soft brake", this approach preserves critical gradient signals from novel discoveries while naturally down-weighting and enabling the reuse of stale, off-policy data. We introduce **$R^2$VPO** (Ratio-Variance Regularized Policy Optimization), which implements this constraint via a primal–dual optimization framework. Extensive evaluations across 7 LLM scales, spanning both fast and slow reasoning paradigms, and 10 robotic control tasks demonstrate the generality of the proposed approach. $R^2$VPO achieves substantial performance gains on mathematical reasoning benchmarks, with particularly pronounced improvements on smaller models, while significantly improving sample efficiency. Furthermore, it consistently outperforms PPO baselines in continuous control domains, particularly in sparse-reward and dynamic environments. Together, these findings establish ratio-variance regularization as a principled foundation for stable and data-efficient policy optimization.

[1]Department of Foundation Model, 2012 Labs, Huawei [2]Shanghai Research Institute for Intelligent Autonomous Systems, Tongji University [3]Department of Computer Science and Technology, Tsinghua University [4]College of Intelligence and Computing, Tianjin University. Correspondence to: Yu Luo <roythu95@gmail.com>, Dong Li <dongleecsu@gmail.com>, Fuchun Sun <fcsun@tsinghua.edu.cn>, Huaping Liu <hpliu@tsinghua.edu.cn>.

*Proceedings of the 43rd International Conference on Machine Learning*, Seoul, South Korea. PMLR 306, 2026. Copyright 2026 by the author(s).

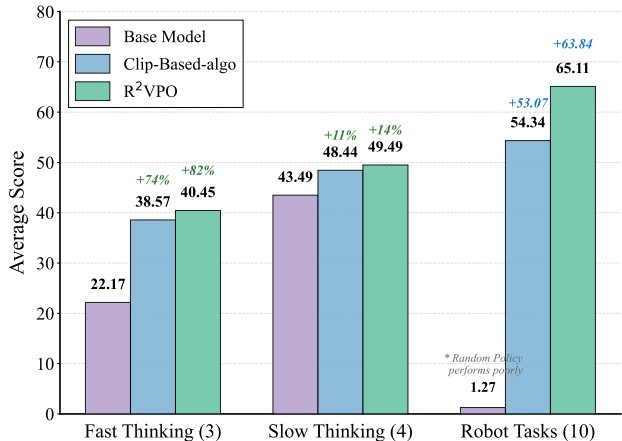

*Figure 1.* **Consistent Average Gains.** Results are aggregated over 5 mathematical reasoning benchmarks across 7 LLM scales (spanning both Fast and Slow thinking paradigms) and 10 continuous robotic control tasks. $R^2$VPO consistently achieves the highest average scores, demonstrating its robustness and superiority in both discrete (LLM) and continuous (Robotics) action spaces.

## 1. Introduction

On-policy reinforcement learning (RL) has become a cornerstone of modern artificial intelligence, underpinning recent advances ranging from the fine-tuning of large language models (LLMs) for alignment and reasoning (Ouyang et al., 2022; Lightman et al., 2023; Shao et al., 2024) to complex locomotion and manipulation in embodied agents (Liu et al., 2025; Lu et al., 2025). By directly optimizing policies against reward signals—whether derived from human feedback, verifiable outcomes, or physical interactions—policy gradient methods provide a flexible and scalable framework for transforming raw predictive models into goal-directed behaviors.

Despite their empirical success, a fundamental challenge persists: ensuring stable and monotonic policy improvement without sacrificing learning efficiency or data utilization. This challenge is traditionally addressed through trust-region methods, most notably Trust Region Policy Optimization (TRPO) (Schulman et al., 2015). While recent works have sought to improve TRPO's scalability via Quasi-Newton methods (Jha et al., 2020), Kronecker-factored approximations (Wu et al., 2017), or low-rank updates (Rozada & Marques, 2023), the computational

burden of second-order optimization remains a barrier. Consequently, first-order approximations such as Proximal Policy Optimization (PPO) (Schulman et al., 2017) and its variants (e.g., GRPO (Shao et al., 2024)) have become widespread. These methods rely on the policy ratio $\rho_t(\theta) = \pi_\theta(a_t|s_t)/\pi_{\text{old}}(a_t|s_t)$ and enforce stability through a heuristic clipping mechanism that truncates $\rho_t$ within a fixed interval.

While clipping has proven effective in practice (Engstrom et al., 2020), it introduces a fundamental limitation: stability is enforced through a pointwise, binary decision. When the policy ratio exceeds a predefined threshold, gradients are abruptly truncated, regardless of the associated return signal. As a result, highly informative samples—such as an LLM discovering a novel reasoning trajectory or a robot executing a previously unseen maneuver—may be suppressed precisely because they induce large policy updates (Xie et al., 2024). Moreover, this rigid boundary renders historical data unusable once it becomes slightly stale, forcing on-policy algorithms to discard valid experiences and rely exclusively on freshly collected samples. Although prior work has proposed various heuristics to relax clipping, including asymmetric bounds or gradient-based modifications (Yu et al., 2025; Su et al., 2025; Roux et al., 2025), these approaches remain fundamentally constrained by the pointwise nature of clipping.

In this work, we argue that this limitation is not inherent to trust-region optimization itself, but rather a consequence of its particular approximation. Revisiting the original trust-region objective, we show that, in the local regime, constraining policy divergence is approximately equivalent to controlling the second-order statistics of the policy ratio—specifically, its variance. This observation reveals variance regularization as a principled, distributional alternative to hard clipping. Unlike pointwise truncation, variance-based constraints softly scale updates according to their divergence, preserving gradient information from high-value outliers while naturally down-weighting stale or off-policy samples. Building on this insight, we propose **Ratio-Variance Regularized Policy Optimization (R$^2$VPO)**. By formulating variance control as a Lagrangian dual constraint, R$^2$VPO employs a primal–dual optimization scheme that jointly maximizes expected returns and minimizes the variance of importance weights, unifying stable on-policy optimization with efficient off-policy data reuse.

We empirically evaluate R$^2$VPO across diverse domains to verify its generality. As summarized in Figure 1, R$^2$VPO demonstrates consistent superiority over strong clipping-based baselines across a wide spectrum of tasks. In the realm of LLM fine-tuning, we conduct extensive experiments on state-of-the-art models across diverse thinking

paradigms. R$^2$VPO achieves a remarkable macro-average relative gain of **+35%** (up to **+138%** on smaller models) and outperforms strong baselines like GRPO in 6 out of 7 settings. Crucially, the off-policy variant (R$^2$VPO-OFF) effectively leverages stale data to improve sample efficiency. Extending our evaluation to continuous control, R$^2$VPO significantly outperforms PPO on DeepMind Control Suite benchmarks (Tassa et al., 2018), exhibiting superior exploration in sparse-reward settings and preventing performance collapse in dynamic tasks. These findings establish ratio-variance control as a highly promising direction for advancing general reinforcement learning [1].

## 2. Related Work

**Approximations of the Trust Region.** Trust region methods ensure monotonic policy improvement by constraining the divergence between consecutive policies (Kakade & Langford, 2002; Schulman et al., 2015), grounded in Natural Gradient descent (Amari, 1998). To address the cost of Fisher Information Matrix computation, various optimizations have been proposed, ranging from K-FAC approximations (ACKTR) (Wu et al., 2017) and Quasi-Newton methods (Jha et al., 2020) to efficient low-rank updates (Rozada & Marques, 2023). Others have focused on theoretical convergence via metric-aware (Song et al., 2023) or stochastic frameworks (Zhao et al., 2019), and extended these principles to multiagent domains (Li & He, 2023). Despite these advances, PPO (Schulman et al., 2017) remains dominant due to its simplicity, utilizing heuristic clipping. Recently, methods like SPO (Xie et al., 2024) and FixPO (Zentner et al., 2023) have revisited KL-regularization, replacing clipping with direct penalties. Similarly, SB-TRPO (Kanwar et al., 2025) incorporates safety bias via trust regions. However, these approaches often rely on fixed coefficients or heuristics. R$^2$VPO advances this direction by deriving the *variance of policy ratio* as a local second-order approximation. Unlike heuristic penalties, our variance-based formulation naturally captures optimization curvature and, by solving the Lagrangian dual, adaptively adjusts constraint strength to avoid hyperparameter sensitivity.

Beyond TRPO/PPO-style approximations, several non-clipping trust-region methods have enforced information-theoretic constraints more directly. Relative Entropy Policy Search (Peters et al., 2010) optimizes policy updates under an explicit KL constraint and solves the resulting dual problem, while subsequent non-parametric policy search methods further study limited information loss under trust-region constraints. V-MPO (Song et al., 2019) formulates policy optimization from a maximum-a-posteriori perspective with KL-constrained E/M-style updates. More recently,

---

[1] We have released our codebase in `https://github.com/Roythuly/R2VPO`

differentiable trust-region layers (Otto et al., 2021) impose constraints by projecting neural policy outputs through differentiable layers. These methods demonstrate that clipping is not the only way to realize trust-region control. Our work is complementary: rather than solving an exact KL-constrained update, introducing projection layers, or estimating Fisher-preconditioned directions, $R^2$VPO extracts the shared local second-order geometry of a broad class of f-divergences and implements it as a simple quadratic regularizer directly in importance-ratio space. This makes it particularly easy to integrate into PPO/GRPO-style LLM and replay-based pipelines.

**On-Policy Optimization in LLMs and Robotics.** PPO has become the de facto standard across domains, dominating robotics tasks (Heess et al., 2017; Hwangbo et al., 2019; Andrychowicz et al., 2020; Zhai et al., 2024; Lu et al., 2025) and LLM alignment (Ouyang et al., 2022; Bai et al., 2022). Recognizing that standard clipping indiscriminately truncates high-value signals, recent heuristics have emerged: DAPO (Yu et al., 2025) and GRPO-CH relax clipping bounds; TOPR (Roux et al., 2025) employs asymmetric thresholds; and GPPO (Su et al., 2025) retains gradient direction. These methods, however, remain "patches" on the clipping paradigm. $R^2$VPO fundamentally differs by enforcing a distributional constraint via variance regulation, providing a soft, adaptive regularization for both the discrete and continuous action spaces.

**Off-Policy Efficiency and Scalability.** In general RL, off-policy algorithms such as SAC (Haarnoja et al., 2018) and TD3 (Fujimoto et al., 2018) achieve superior sample efficiency but require separate Critic networks, which poses scalability hurdles for 7B+ models. While works like Off-Policy TRPO (Meng et al., 2021; Li, 2023) have introduced monotonic improvement guarantees for off-policy reuse, adapting them to critic-free LLM fine-tuning remains challenging. Alternative approaches using Importance Sampling (IS) corrections (e.g., V-trace (Espeholt et al., 2018), Retrace (Munos et al., 2016)) often suffer from high variance with stale data. $R^2$VPO circumvents these bottlenecks by leveraging ratio variance as a natural stabilizer. By penalizing the variance of importance weights, our framework enables robust off-policy reuse without complex correction terms or extra Value networks, combining the efficiency of methods like Off-Policy TRPO with the scalability of critic-free architectures.

## 3. Preliminaries

We consider a standard RL framework modeled as a Markov Decision Process (MDP), defined by the tuple $\mathcal{M} = \langle \mathcal{S}, \mathcal{A}, \mathcal{P}, r, \gamma \rangle$. Here, $\mathcal{S}$ denotes the state space, $\mathcal{A}$ is the action space, and $\mathcal{P} : \mathcal{S} \times \mathcal{A} \to \Delta(\mathcal{S})$ represents

the transition dynamics. At each time step $t$, the agent observes a state $s_t \in \mathcal{S}$, selects an action $a_t \in \mathcal{A}$ according to a stochastic policy $\pi_\theta(a_t|s_t)$ parameterized by $\theta$, receives a scalar reward $r(s_t, a_t)$, and transitions to the next state $s_{t+1} \sim \mathcal{P}(\cdot|s_t, a_t)$. The discount factor $\gamma \in [0, 1)$ balances immediate and future rewards. The standard goal in RL is to find an optimal policy $\pi^*$ that maximizes the expected discounted cumulative return

$$\mathcal{J}(\pi_\theta) = \mathbb{E}_{\tau \sim \pi_\theta}\left[\sum_{t=0}^{\infty} \gamma^t r(s_t, a_t)\right], \qquad (1)$$

where $\tau = (s_0, a_0, s_1, a_1, \dots)$ denotes a trajectory sampled under the policy $\pi_\theta$.

Optimizing $\mathcal{J}(\pi_\theta)$ directly via vanilla policy gradients often suffers from instability due to large step sizes. To ensure monotonic improvement, Trust Region Policy Optimization (TRPO) (Schulman et al., 2015) maximizes a surrogate objective subject to a strict KL-divergence constraint on the policy update, which guarantees policy improvement provided the new policy stays within a "trust region" of the old policy

$$\begin{aligned} \max_\theta \quad & \mathbb{E}_t\left[\rho_t(\theta)\,\hat{A}_t\right] \\ \text{s.t.} \quad & \mathbb{E}_t\left[D_{\text{KL}}\left(\pi_{\text{old}}(\cdot|s_t)\,\|\,\pi_\theta(\cdot|s_t)\right)\right] \leq \delta \end{aligned} \qquad (2)$$

where $\rho_t(\theta) = \frac{\pi_\theta(a_t|s_t)}{\pi_{\text{old}}(a_t|s_t)}$ is the probability ratio, $\hat{A}_t$ is the advantage function estimator, and $\delta$ is a divergence limit. Although expressed in terms of KL divergence, the trust-region constraint implicitly enforces that the policy ratio remains close to one over the data distribution.

Due to the high computational cost of enforcing Eq. (2) via second-order optimization, first-order methods such as Proximal Policy Optimization (PPO) (Schulman et al., 2017) and Group Relative Policy Optimization (GRPO) (Shao et al., 2024) have become standard. These algorithms replace the explicit trust-region constraint with a heuristic clipping objective:

$$\begin{aligned} \mathcal{L}^{\text{CLIP}}(\theta) = \mathbb{E}_t\Big[ \min\Big( & \rho_t(\theta)\,\hat{A}_t, \\ & \text{clip}(\rho_t(\theta), 1 - \epsilon, 1 + \epsilon)\,\hat{A}_t\Big)\Big]. \end{aligned} \qquad (3)$$

Clipping enforces stability by imposing a fixed, pointwise bound on the policy ratio. While effective at preventing destructive updates, this binary mechanism tightly couples stability to the hard truncation of $\rho_t(\theta)$. As a result, it indiscriminately discards gradient information from samples with high divergence or staleness, limiting both learning efficiency and the reuse of off-policy data. This limitation motivates the need for a more principled regularization that controls divergence broadly without erasing valuable high-leverage signals.

*Table 1.* **Quadratic approximation of common $f$-divergences in the trust-region regime** ($\rho_\theta \to 1$). All approximations follow the second-order expansion: $D_f(\pi_\theta \| \pi_{\text{off}}) \approx \frac{f''(1)}{2} \operatorname{Var}_{\pi_{\text{off}}}[\rho_\theta]$.

| Divergence | Generator $f(u)$ | $f''(1)$ | Variance Approximation |
|---|---|---|---|
| Reverse KL | $u \log u$ | 1 | $D_{\text{RKL}}(\pi_\theta \| \pi_{\text{off}}) \approx \frac{1}{2} \operatorname{Var}[\rho_\theta]$ |
| Forward KL | $-\log u$ | 1 | $D_{\text{FKL}}(\pi_\theta \| \pi_{\text{off}}) \approx \frac{1}{2} \operatorname{Var}[\rho_\theta]$ |
| Jensen-Shannon | $\frac{u}{2} \log u - \frac{u+1}{2} \log \frac{u+1}{2}$ | 1/4 | $D_{\text{JS}}(\pi_\theta \| \pi_{\text{off}}) \approx \frac{1}{8} \operatorname{Var}[\rho_\theta]$ |
| Hellinger | $(\sqrt{u} - 1)^2$ | 1/2 | $D_{\text{H}}(\pi_\theta \| \pi_{\text{off}}) \approx \frac{1}{4} \operatorname{Var}[\rho_\theta]$ |
| $\chi^2$-divergence | $(u - 1)^2$ | 2 | $D_{\chi^2}(\pi_\theta \| \pi_{\text{off}}) \approx \operatorname{Var}[\rho_\theta]$ |
| $\alpha$-divergence ($\alpha = 0.5$) | $4(1 - \sqrt{u})$ | 1 | $D_\alpha(\pi_\theta \| \pi_{\text{off}}) \approx \frac{1}{2} \operatorname{Var}[\rho_\theta]$ |

## 4. Method

In this section, we revisit trust-region policy optimization from the perspective of the *policy ratio*. We show that, across a broad class of trust-region constraints, the local geometry of policy divergence is universally governed by the variance of the policy ratio. This insight leads to a simple yet principled alternative to heuristic clipping: a variance-regularized objective that enables stable updates and naturally supports off-policy data reuse.

### 4.1. From $f$-Divergence to Policy Ratio Variance

To establish a generalized framework, we revisit the trust region constraint. While TRPO specifically employs the Kullback-Leibler (KL) divergence, we consider the broader family of $f$-divergences (Sason & Verdú, 2016), denoted as $D_f(\cdot \| \cdot)$. Let $\pi_{\text{off}}$ be the behavior policy (or the policy at the previous iteration) and $\pi_\theta$ be the current policy. The trust region constraint can be formulated as

$$\mathbb{E}_{s \sim d^{\pi_{\text{off}}}} \left[ D_f(\pi_\theta(\cdot|s) \| \pi_{\text{off}}(\cdot|s)) \right] \leq \delta, \tag{4}$$

where the divergence is defined as $D_f(P \| Q) = \mathbb{E}_{x \sim Q}[f(P(x)/Q(x))]$ and $f : (0, \infty) \to \mathbb{R}$ is a convex function with $f(1) = 0$. While valid for any convex $f$, directly optimizing this constraint is often intractable. Here, we demonstrate that for any valid $f$-divergence, the local geometry of the trust region is dominated by the variance of the policy ratio.

**Proposition 4.1** (Variance Approximation of $f$-Divergence). *Let $\pi_\theta$ be a policy in the neighborhood of a reference policy $\pi_{\text{off}}$, and define the policy ratio $\rho_\theta(a|s) = \pi_\theta(a|s)/\pi_{\text{off}}(a|s)$. Assume $f$ is twice continuously differentiable at $\rho = 1$ with $f(1) = 0$. Then, for sufficiently small policy updates (i.e., $\rho_\theta \to 1$), the expected $f$-divergence admits the second-order approximation*

$$\mathbb{E}_{\pi_{\text{off}}}[D_f(\pi_\theta \| \pi_{\text{off}})] = \frac{f''(1)}{2} \operatorname{Var}_{\pi_{\text{off}}}[\rho_\theta] + \mathcal{O}\left(\mathbb{E}_{\pi_{\text{off}}}[|\rho_\theta - 1|^3]\right). \tag{5}$$

*where the variance is taken over $(s, a) \sim \pi_{\text{off}}$.*

The proof is provided in Appendix A.1. This high-order term indicates that the approximation is accurate in a small neighborhood around $\rho_\theta \approx 1$. Thus, ratio variance constitutes the fundamental second-order surrogate underlying a wide class of trust-region constraints.

*Remark* 4.2. While Proposition 4.1 is derived as a local approximation around $\rho \approx 1$, variance regularization remains a conservative stabilizer even beyond this regime, as it directly penalizes the second moment of importance weights. Unlike hard clipping, which introduces a discontinuous truncation, the quadratic penalty grows smoothly and provides meaningful gradients for large deviations.

As summarized in Table 1, this property holds uniformly across the $f$-divergence family. In the local trust-region regime, common divergences—including Forward/Reverse KL, Jensen–Shannon, Hellinger, and $\chi^2$—all collapse to a scaled ratio-variance term $\frac{f''(1)}{2} \operatorname{Var}[\rho_\theta]$. Figure 2 empirically validates this approximation, showing tight alignment between exact divergences and their quadratic surrogates across representative metrics. Additional visualizations are provided in Appendix B.

### 4.2. Primal-Dual Formulation

Leveraging the approximation established in Prop. 4.1, we recast the intractable $f$-divergence constrained problem into a tractable variance-constrained optimization. By substituting the variance term $\mathbb{E}[(\rho_t(\theta) - 1)^2]$ for the divergence constraint, the optimization objective becomes

$$\max_\theta \quad \mathbb{E}_{(s_t, a_t) \sim \pi_{\text{off}}} \left[ \rho_t(\theta) \hat{A}_t \right]$$
$$\text{s.t.} \quad \mathbb{E}_{(s_t, a_t) \sim \pi_{\text{off}}} \left[ (\rho_t(\theta) - 1)^2 \right] \leq \delta. \tag{6}$$

Solving this constrained problem directly is challenging due to the hard boundary on the variance. To address this, we employ the method of Lagrange multipliers to relax the constraint into a penalty term, transforming the problem into an unconstrained saddle-point optimization.

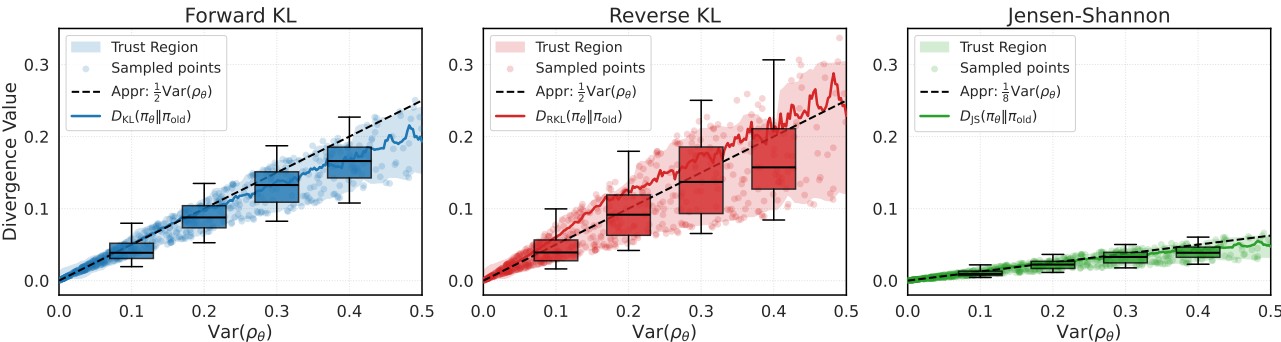

*Figure 2.* **Ratio-variance as a unified proxy for $f$-divergence trust regions**. Theoretical quadratic approximations (dashed lines) align tightly with exact numerical values (solid lines) across Reverse KL, Forward KL, and JS metrics. Boxplots and shading represent 80% confidence intervals from sampled Gaussian policies, confirming that ratio variance provides a stable, computationally tractable alternative to complex divergence constraints.

**Theorem 4.3** ($\mathrm{R}^2\mathrm{VPO}$ Primal-Dual Objective). *The constrained optimization problem in Eq. (6) can be written as the following Lagrangian min–max problem:*

$$\min_{\lambda \geq 0} \max_{\theta} \mathcal{L}(\theta, \lambda) = \mathbb{E}_{\pi_{\mathit{off}}} \left[ \rho_t(\theta)\hat{A}_t - \lambda \left( (\rho_t(\theta) - 1)^2 - \delta \right) \right],$$
(7)

*where $\lambda$ is the dual variable (Lagrange multiplier) dynamically controlling the strength of the variance regularization, and $\delta$ represents the target tolerance level.*

To understand the mechanism of this objective, we analyze the gradient of the Lagrangian $\mathcal{L}$ with respect to the policy

$$\nabla_\theta \mathcal{L}(\theta, \lambda) = \mathbb{E}_{(s_t, a_t) \sim \pi_{\mathrm{off}}} \left[ \underbrace{\left( \hat{A}_t - 2\lambda(\rho_t(\theta) - 1) \right)}_{\text{Regularized Advantage}} \rho_t(\theta) \right.$$
$$\left. \times \nabla_\theta \log \pi_\theta(a_t \mid s_t) \right].$$
(8)

The gradient in Eq. (8) reveals the core mechanism of $\mathrm{R}^2\mathrm{VPO}$. The term $2\lambda(\rho_t(\theta)-1)$ acts as a dynamic, instance-dependent regularizer that modulates the original advantage $\hat{A}_t$. Unlike hard clipping, which zeroes out the gradient when the ratio exceeds a threshold $\nabla\mathcal{L} = 0$, this formulation preserves the gradient direction but scales its magnitude:

- **Near-Identity Updates:** When the policy ratio is close to 1 (i.e., $\pi_\theta \approx \pi_{\mathrm{off}}$), the penalty term vanishes, and the update is driven purely by the task advantage $\hat{A}_t$.

- **High-Divergence Updates:** When the policy deviates significantly, the penalty term grows linearly with the deviation. This induces a *continuous attenuation* of high-divergence updates, in contrast to the discontinuous truncation imposed by clipping.

To rigorously justify substituting the established hard clipping mechanism with our variance-based control, we derive

a bound relating the two approaches. We show that the "clipping error"—the divergence between the true off-policy objective and the heuristically clipped surrogate—is strictly upper-bounded by the policy ratio variance.

**Theorem 4.4** (Variance Bound on Clipping Error). *Let $\mathcal{J}^{\mathrm{UNC}}(\theta) = \mathbb{E}_{(s_t, a_t) \sim \pi_{\mathrm{off}}}[\rho_t(\theta)\hat{A}_t]$ be the unclipped off-policy objective, and $\mathcal{J}^{\mathrm{CLIP}}(\theta)$ the clipped objective with clip range $\epsilon$. Assume that the support of $\pi_\theta$ is contained in that of $\pi_{\mathrm{off}}$, and that $\mathbb{E}_{\pi_{\mathrm{off}}}[\rho_t^2] < \infty$. Then,*

$$\left| \mathcal{J}^{\mathrm{UNC}}(\theta) - \mathcal{J}^{\mathrm{CLIP}}(\theta) \right| \leq \frac{A_{\max}}{\epsilon} \mathrm{Var}_{\pi_{\mathrm{off}}}[\rho_t(\theta)]. \quad (9)$$

The proof is provided in Appendix A.2. Theorem 4.4 offers a formal justification for $\mathrm{R}^2\mathrm{VPO}$. By explicitly minimizing the ratio variance (via the penalty $\lambda$), $\mathrm{R}^2\mathrm{VPO}$ naturally minimizes the potential overestimation error that clipping was heuristically designed to prevent. Unlike hard clipping, which completely discards signals outside the $\epsilon$-bound, variance control softly penalizes them proportional to their squared deviation, maintaining gradient flow while reducing the clipping-induced error controlled by the ratio variance.

$\mathrm{R}^2\mathrm{VPO}$ should be interpreted as a local trust-region surrogate rather than an exact trust-region method with global monotonic improvement guarantees. Unlike TRPO, which enforces an explicit KL constraint and derives a monotonic improvement bound under idealized assumptions, our method replaces the exact divergence by its local second-order ratio-variance approximation. Therefore, the guarantee provided by Proposition 4.1 is local: when the policy ratio remains close to one, the variance penalty captures the leading-order geometry of the trust region. In exchange for giving up exact global monotonicity, $\mathrm{R}^2\mathrm{VPO}$ obtains a simple first-order objective that operates directly in the ratio space used by PPO/GRPO-style training and naturally supports stale-data reuse.

---

**Algorithm 1** Training Frameworks for R$^2$VPO

---

| **Algorithm 1** R$^2$VPO-ON (On-Policy) | **Algorithm 2** R$^2$VPO-OFF (Off-Policy) |
|---|---|
| **Require:** Initial policy $\pi_\theta$, constraint $\delta$, learning rate $\eta_\lambda$ | **Require:** Policy $\pi_\theta$, Replay Buffer $\mathcal{D}$, constraint $\delta$, rate $\eta_\lambda$ |
| 1: Initialize dual variable $\lambda \leftarrow 0.04$ | 1: Initialize dual variable $\lambda \leftarrow 0.04$ |
| 2: **repeat** | 2: **repeat** |
| 3:    *// 1. Online Data Collection* | 3:    *// 1. Inference & Data Storage* |
| 4:    Collect trajectories $\tau \sim \pi_\theta$ | 4:    **for** each query batch $x$ **do** |
| 5:    Compute rewards $r$ by verifier or reward model | 5:      Sample $y \sim \pi_\theta(\cdot|x)$, compute $r, \hat{A}$ |
| 6:    Compute advantages $\hat{A}$ by GAE or relative group | 6:      Store tuple $\langle x, y, r, \hat{A}, \log \pi_{\text{off}} \rangle$ in $\mathcal{D}$ |
| 7:    *// 2. Primal-Dual Optimization* | 7:    **end for** |
| 8:    **for** epoch $k = 1, \ldots, K$ **do** | 8:    *// 2. Primal-Dual Optimization* |
| 9:      Compute ratio $\rho_t$ | 9:    **for** gradient step $j = 1, \ldots, M$ **do** |
| 10:     Compute loss $\mathcal{L}(\theta, \lambda)$ per Eq. (7) | 10:     Sample batch $b \sim \mathcal{D}$ |
| 11:     Update primal: $\theta \leftarrow \theta + \alpha \nabla_\theta \mathcal{L}$ | 11:     Compute ratio $\rho_t$ (using stored $\log \pi_{\text{off}}$) |
| 12:     *// (Optional) Dual Update* | 12:     Compute loss $\mathcal{L}(\theta, \lambda)$ on batch $b$ per Eq. (7) |
| 13:     Update $\lambda$ via dual descent using Eq. (10) | 13:     Update primal: $\theta \leftarrow \theta + \alpha \nabla_\theta \mathcal{L}$ |
| 14:    **end for** | 14:     *// (Optional) Dual Update* |
| 15: **until** max environment steps | 15:     Update $\lambda$ via dual descent using Eq. (10) |
| | 16:    **end for** |
| | 17: **until** max environment steps |

---

## 4.3. Implementation Instances

The theoretical framework developed in Theorem 4.3 directly gives rise to **R$^2$VPO**, a practical algorithm that replaces heuristic clipping with a principled variance-based regularization. Concretely, the clip operator and its associated thresholds are substituted by a simple quadratic penalty $(\rho_t(\theta) - 1)^2$, yielding a clean objective that is fully aligned with the primal–dual formulation in Eq. (7). This design not only simplifies implementation but also provides a unified interface for both on-policy and off-policy training, as summarized in Algorithm 1.

**On-Policy Training (R$^2$VPO-ON).** In the standard on-policy setting, R$^2$VPO mirrors the workflow of PPO/GRPO but employs the Lagrangian objective. A flexibility of our framework lies in the option of the dual variable $\lambda$, which governs the regularization strength. It can be configured in two modes depending on stability requirements:

1. **Fixed Regularization:** $\lambda$ is treated as a static hyperparameter (e.g., $\lambda = 0.04$) to simplify implementation.

2. **Adaptive Lagrangian Tuning:** $\lambda$ is dynamically optimized via dual gradient descent to strictly enforce the trust region constraint $\delta$.

In the adaptive setting, $\lambda$ is updated at each step via

$$\lambda \leftarrow \max(0, \lambda - \eta_\lambda(\delta - \mathbb{E}[(\rho_t - 1)^2])). \qquad (10)$$

This automatic tuning ensures the policy remains within the trust region without manual hyperparameter searching.

**Off-Policy Training (R$^2$VPO-OFF).** Crucially, the variance-based formulation extends naturally to off-policy learning by robustly tolerating data staleness. Unlike standard on-policy methods that must discard data immediately after optimization, R$^2$VPO-OFF leverages a *Replay Buffer* $\mathcal{D}$ to reuse historical reasoning trails. During the inference phase, we collect experience tuples $\tau = \langle q, o, \log \pi_{\text{off}}(o \mid q), r, \hat{A} \rangle$ and store them in $\mathcal{D}$, explicitly recording the behavior policy $\pi_{\text{off}}$ at generation time. And in the training phase, we sample mini-batches uniformly from $\mathcal{D}$. The variance penalty $(\rho_t - 1)^2$ automatically scales the gradient contribution: fresh data contributes fully, while stale data is softly penalized via the "Regularized Advantage" mechanism, preventing instability. Similar to the on-policy variant, the regularization coefficient $\lambda$ can be either fixed or adaptively tuned. Beyond sample efficiency, this off-policy capability facilitates a decoupled, asynchronous architecture ideal for LLM post-training. Since generation (inference) and optimization (training) often have mismatched throughputs, R$^2$VPO-OFF allows Actor nodes to focus exclusively on high-throughput generation while Learner nodes asynchronously update the model using buffered data. The updated weights are periodically synchronized to Actors, maximizing overall system efficiency.

## 5. Experiment

In this section, we conduct a comprehensive evaluation to verify the efficacy of R$^2$VPO. Our experiments are designed to answer two primary research questions (RQ):

- **RQ1 (Asymptotic Performance):** Does ratio variance regularization provide a more stable and effective optimization landscape than heuristic hard clipping, leading to superior reasoning capabilities?

- **RQ2 (Sample Efficiency):** Can R$^2$VPO effectively

*Table 2.* **Streamlined Performance Comparison.** By merging the accuracy score and relative gain into a single cell, we provide a clearer view of performance across all seven model scales. Values are presented as **Accuracy** (Gain). $R^2$VPO-OFF consistently achieves the highest relative gains across diverse thinking paradigms.

| Method | Fast Thinking Models | | | Slow Thinking Models | | | | Summary |
|---|---|---|---|---|---|---|---|---|
| | **Pangu-1B** | **Qwen-1.7B** | **Qwen-8B** | **DS-1.5B** | **Pangu-7B** | **Qwen-4B** | **Qwen-8B** | **Macro-Avg** |
| *Base Model* | 23.22 | 15.95 | 27.33 | 29.52 | 41.60 | 51.60 | 51.22 | 34.35 |
| GRPO | 27.03 (+16%) | 30.70 (+93%) | 52.53 (+92%) | 29.23 (-1.0%) | 54.98 (+32%) | 50.13 (-2.8%) | 52.93 (+3.4%) | 42.50 (+24%) |
| GRPO-CH | 28.02 (+21%) | 33.78 (+112%) | 55.08 (+102%) | 30.87 (+4.6%) | 57.50 (+38%) | 52.15 (+1.1%) | 56.70 (+11%) | 44.87 (+31%) |
| GPPO | 27.18 (+17%) | 36.43 (+128%) | 55.75 (+104%) | 31.05 (+5.2%) | 56.09 (+35%) | 52.00 (+0.8%) | 54.72 (+6.8%) | 44.75 (+30%) |
| TOPR | 26.67 (+15%) | 34.68 (+117%) | 55.03 (+101%) | 31.21 (+5.7%) | 55.60 (+34%) | **53.13** (+3.0%) | 55.67 (+8.7%) | 44.57 (+30%) |
| **$R^2$VPO-ON** | 27.78 (+20%) | 37.37 (+134%) | 53.93 (+97%) | 29.45 (-0.2%) | **57.55** (+38%) | 52.93 (+2.6%) | 55.30 (+8.0%) | 44.90 (+31%) |
| **$R^2$VPO-OFF** | **28.17** (+21%) | **38.03** (+138%) | **57.40** (+110%) | **34.17** (+16%) | 56.32 (+35%) | 52.00 (+0.8%) | **58.20** (+14%) | **46.33** (+35%) |

leverage off-policy data via replay buffers to significantly reduce the computational cost (rollouts) required for convergence?

To strictly address these questions, our experimental design spans the full spectrum of action modalities. We first evaluate $R^2$VPO in the high-dimensional discrete action space of LLMs, specifically on complex mathematical reasoning benchmarks, to rigorously test its exploration capabilities and data reuse efficiency. Complementing this, we extend our assessment to the continuous action space of robotic locomotion and manipulation to verify the algorithm's robustness and physical stability.

For LLM reasoning, models are fine-tuned with a global batch size of 128 and learning rate of $1 \times 10^{-6}$, utilizing an *adaptive* dual-update strategy for $\lambda$. We evaluate both on-policy ($R^2$VPO-ON) and off-policy ($R^2$VPO-OFF) regimes; the latter employs a FIFO replay buffer (capacity 4) with an Update-to-Data (UTD) ratio of 2 to efficiently leverage historical data. For continuous control, agents utilize MLP networks as backend. We leverage massive parallelism ($2,048$ concurrent environments) for robust gradient estimation. Unlike the LLM setting, we employ a *fixed* dual factor $\lambda$ given the dense reward signals in robotic tasks for training stability. Full hyperparameter configurations are detailed in Appendix C.

### 5.1. Complex Reasoning with LLMs

We utilize the DAPO-Math-17K dataset (Yu et al., 2025) for training. To assess generalization, we evaluate performance on five challenging mathematical reasoning benchmarks: AIME 2024, AIME 2025 (Committees, 2024,2025), AMC 2023 (AI-MO, 2023), HMMT Feb 2025 (Tournament, 2025), and OlymMath (Sun et al., 2025). Our experiments cover a comprehensive set of seven base models spanning both *Fast Thinking* (e.g., Qwen3-1.7B/8B-Fast (Yang et al.,

2025), openPangu-1B (Chen et al., 2025)) and *Slow Thinking* paradigms (e.g., DeepSeek-Distill-Qwen-1.5B (Shao et al., 2024), openPangu-7B (Chen et al., 2025), Qwen3-4/8B-Thinking (Yang et al., 2025))[2]. For comparative analysis, we benchmark $R^2$VPO against four state-of-the-art policy gradient methods: GRPO (Shao et al., 2024), which employs standard symmetric clipping; GRPO-CH (Yu et al., 2025), a variant with relaxed upper clipping thresholds; GPPO (Su et al., 2025), which prioritizes gradient direction preservation; and TOPR (Roux et al., 2025), which utilizes asymmetric estimators for off-policy corrections.

Table 2 demonstrates the consistent superiority of $R^2$VPO. **$R^2$VPO-OFF** achieves the highest macro-average accuracy of **46.33%** (**+35%** relative gain over base models), significantly outperforming strong baselines like GRPO (+24%) and GPPO (+30%). The gains are most dramatic in Fast Thinking models, where *Qwen3-1.7B-Fast* and *Qwen3-8B-Fast* improve by **138%** and **110%**, respectively. As analyzed in Appendix D.1, these performance jumps coincide with a substantial increase in response length, suggesting that variance-aware optimization successfully unlocks extended reasoning capabilities in smaller models. Furthermore, $R^2$VPO-OFF outperforms its on-policy counterpart in 5 out of 7 settings, confirming that variance regularization effectively mitigates the instability of stale data. And, $R^2$VPO-ON remains highly competitive, notably surpassing the off-policy variant on *openPangu-7B* (57.55%), which validates the stability of the core variance-regularized objective even without data replay.

---

[2]Following the Qwen3 technical report (Yang et al., 2025), we explicitly control the reasoning paradigm via prompting: appending `<think>` triggers the 'Slow Thinking' mode (long-chain reasoning), while appending `<think></think>` enforces the 'Fast Thinking' mode (direct response).

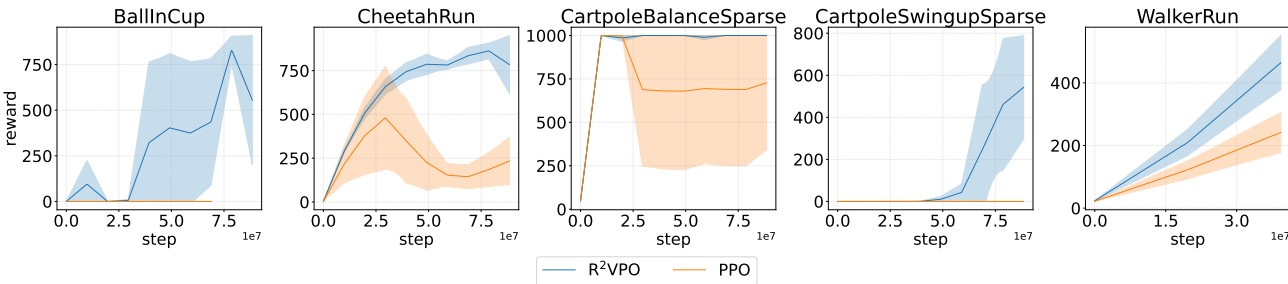

*Figure 3.* **Training Curves on Continuous Control Tasks (DeepMind Control Suite).** We compare $R^2$VPO-ON (blue) against PPO (orange) across locomotion and manipulation tasks, including those with sparse rewards. Solid lines denote the mean performance over 5 independent training runs with different random seeds, and shaded regions denote standard deviation. $R^2$VPO demonstrates superior exploration in sparse settings (e.g., `Ball_in_Cup`) and prevents the performance collapse observed in PPO (e.g., `Cheetah_Run`), highlighting the robustness of variance-aware constraints.

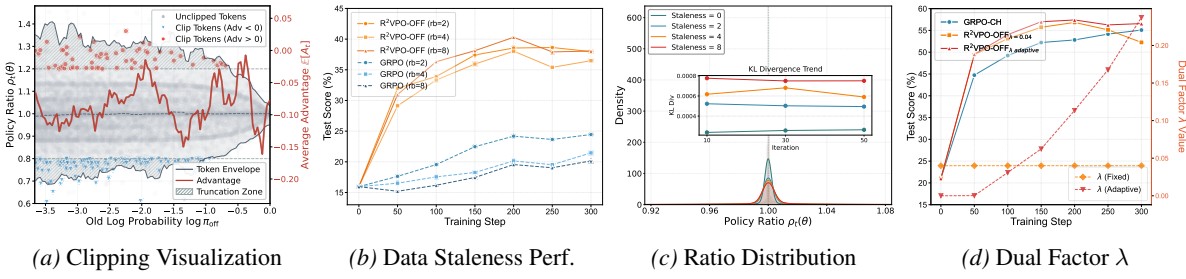

*(a)* Clipping Visualization   *(b)* Data Staleness Perf.   *(c)* Ratio Distribution   *(d)* Dual Factor $\lambda$

*Figure 4.* **Mechanism Analysis. (a)** Hard clipping indiscriminately truncates high-value exploration. **(b)** $R^2$VPO exhibits superior robustness to data staleness compared to GRPO. **(c)** Bounded ratio distributions empirically validate the reliability of the second-order approximation. **(d)** The adaptive dual-update strategy ($\lambda_{\text{adaptive}}$) outperforms fixed constraints.

## 5.2. Continuous Robotic Control

To demonstrate generality beyond discrete reasoning, we evaluate $R^2$VPO in on-policy manner on continuous control tasks from DM control (Tassa et al., 2018), covering locomotion (e.g., `Walker`, `Cheetah`) and manipulation (e.g., `Ball_in_Cup`). As illustrated in Figure 3, $R^2$VPO exhibits significantly improved robustness compared to the PPO baseline. In sparse-reward tasks like `Ball_in_Cup`, PPO fails to learn (flatlining near zero), whereas $R^2$VPO successfully solves the task, evidencing superior exploration. In dense-reward tasks like `Cheetah_Run`, PPO suffers from catastrophic performance collapse during late-stage training. In contrast, $R^2$VPO exhibits steady improvement without the late-stage collapse observed in PPO, suggesting that explicitly controlling the ratio variance yields a more stable optimization landscape than heuristic clipping. Full results on 10 environments are provided in Appendix D.2.

## 5.3. Analysis of $R^2$VPO under Ratio

**The Necessity of Variance-Aware Constraints.** Figure 4(a) visualizes a critical conflict in standard methods: high-value exploration tokens (red points) naturally exhibit high variance and often fall into the clipping region. Standard mechanisms indiscriminately zero out gradients for these informative samples. $R^2$VPO addresses this by re-

placing binary clipping with a distributional "soft brake". This design is validated by the dynamics of the adaptive dual variable $\lambda$ shown in Figure 4(c). During early training, $\lambda$ remains naturally inactive ($\approx 0$), effectively simplifying the objective to an unconstrained form. This implies that the strict clipping enforced by GRPO during early phases is unnecessary and potentially detrimental. As the policy diverges later in training, $\lambda$ gradually increases (red dashed line), applying a stricter penalty exactly when needed to ensure stability. Consequently, the adaptive $\lambda$ strategy yields superior asymptotic performance compared to both fixed-$\lambda$ and GRPO-CH baselines.

**Robustness to Data Staleness.** We further investigate the impact of off-policy data staleness by varying the replay buffer capacity ($rb \in \{2, 4, 8\}$) on the *Qwen3-1.7B-Fast* model. As shown in the performance curves in Figure 4(b), GRPO is highly sensitive to staleness, exhibiting significant degradation as the buffer size increases. In sharp contrast, $R^2$VPO demonstrates remarkable robustness. To understand the mechanism behind this, Figure 4(c) visualizes the distribution of policy ratios $\rho_t$ under varying staleness. We observe that in the context of LLM fine-tuning, increasing staleness does not lead to excessively extreme ratio distributions; this bounded behavior provides empirical support for the reliability of the second-order Taylor expansion un-

derlying our variance-based constraint, even in off-policy regimes. $R^2$VPO effectively tolerates the resulting distribution shifts (as shown by the evolving average ratio in the inset plot) throughout training. This confirms that the variance-based penalty allows for the safe handling of stale data without the destructive truncation inherent in hard clipping. Additional visualizations of ratio dynamics across training steps are provided in Appendix D.1.

**Ablations on the dual factor $\lambda$.** To validate the effectiveness of our adaptive constraint mechanism, we compare two distinct strategies on the *Qwen3-8B-Fast* model: a static assignment ($\lambda = 0.04$) and our proposed dynamic dual-update mechanism ($\lambda_{\text{adaptive}}$). As shown in Figure 4(c), both variance-regularized variants consistently outperform the GRPO-CH baseline on all benchmarks, confirming the fundamental efficacy of the ratio variance penalty. Crucially, the adaptive variant yields superior asymptotic performance over the static one. The visualization of $\lambda_{\text{adaptive}}$ (red dashed line) reveals a distinct trajectory: it remains near zero during the early "warm-up" phase and gradually increases as training progresses. This confirms our hypothesis that while early exploration benefits from relaxed constraints, a stricter penalty is increasingly necessary to stabilize updates as the policy diverges from the behavior distribution.

## 6. Conclusion

In this work, we proposed $R^2$VPO, a principled framework that transcends the limitations of heuristic hard clipping in general RL. By theoretically establishing ratio variance as a superior, distributional proxy for the trust region, we shifted the paradigm from indiscriminate "pointwise truncation" to adaptive "soft regularization", which preserves gradient signals from high-divergence yet informative exploration, and naturally accommodates the distribution shifts inherent in off-policy learning. Empirical evaluations across two distinct domains confirm that $R^2$VPO significantly outperforms standard clipping-based baselines in both asymptotic performance and sample efficiency. While variance regularization provides a robust approximation of the trust region, the penalty term can disproportionately dominate the optimization objective in regimes of extreme stochasticity or large policy divergence. This may result in overly conservative updates that slow down convergence in early training phases. Future work will investigate adaptive scaling mechanisms to mitigate the impact of excessive variance and extend this framework to complex, multi-turn Agentic RL workflows, where stability in long-horizon planning is critical.

## Acknowledgements

We sincerely thank all co-authors for their valuable contributions, insightful discussions, and continuous support throughout this work. We are also grateful to the reviewers for their constructive comments and suggestions, which helped us improve the clarity, positioning, and overall quality of the paper.

## Impact Statement

This paper introduces $R^2$VPO, a framework aimed at enhancing the stability and sample efficiency of reinforcement learning in both Large Language Models (LLMs) and robotic control. The primary societal impact of this work lies in its potential to significantly reduce the computational resources required for training capable AI systems. By achieving convergence with approximately fewer rollouts and enabling efficient off-policy data reuse, our method contributes to the goals of "Green AI", lowering the carbon footprint associated with large-scale model post-training.

Furthermore, by improving the training stability of continuous control policies, this work supports the development of safer and more reliable embodied agents, reducing the risks of physical accidents during deployment. However, as with all advancements that enhance the reasoning and control capabilities of general-purpose AI, there remains a risk of misuse if applied to optimize harmful objectives. We advocate for the responsible deployment of these algorithms, accompanied by robust safety alignment protocols.

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

## A. Theoretical Analyses

**Proposition A.1** (Variance Approximation of $f$-Divergence)**.** *Let $\pi_\theta$ be a policy in the neighborhood of a reference policy $\pi_{\text{off}}$, and define the policy ratio $\rho_\theta(a|s) = \pi_\theta(a|s)/\pi_{\text{off}}(a|s)$. Assume $f$ is twice continuously differentiable at $\rho = 1$ with $f(1) = 0$. Then, for sufficiently small policy updates (i.e., $\rho_\theta \to 1$), the expected $f$-divergence admits the second-order approximation*

$$\mathbb{E}_{\pi_{\text{off}}}[D_f(\pi_\theta \| \pi_{\text{off}})] = \frac{f''(1)}{2} \text{Var}_{\pi_{\text{off}}}[\rho_\theta] + \mathcal{O}\left(\mathbb{E}_{\pi_{\text{off}}}[|\rho_\theta - 1|^3]\right). \tag{11}$$

*where the variance is taken over $(s, a) \sim \pi_{\text{off}}$.*

*Proof.* By definition, the $f$-divergence between $\pi_\theta$ and $\pi_{\text{off}}$ can be written as an expectation under the reference policy:

$$D_f(\pi_\theta \| \pi_{\text{off}}) = \mathbb{E}_{(s,a)\sim\pi_{\text{off}}}[f(\rho_\theta(a|s))], \tag{12}$$

where $\rho_\theta(a|s) = \pi_\theta(a|s)/\pi_{\text{off}}(a|s)$. Since $\pi_\theta$ is assumed to be close to $\pi_{\text{off}}$, the ratio $\rho_\theta$ concentrates around 1. We perform a second-order Taylor expansion of $f(\rho)$ around $\rho = 1$:

$$f(\rho) = f(1) + f'(1)(\rho - 1) + \frac{1}{2}f''(1)(\rho - 1)^2 + \mathcal{O}(|\rho - 1|^3). \tag{13}$$

Substituting this expansion into the definition of the divergence and taking expectation under $\pi_{\text{off}}$, we obtain

$$\mathbb{E}_{\pi_{\text{off}}}[f(\rho_\theta)] = f(1) + f'(1)\mathbb{E}_{\pi_{\text{off}}}[\rho_\theta - 1] + \frac{1}{2}f''(1)\mathbb{E}_{\pi_{\text{off}}}[(\rho_\theta - 1)^2] + \mathcal{O}\left(\mathbb{E}_{\pi_{\text{off}}}[|\rho_\theta - 1|^3]\right). \tag{14}$$

By construction of the policy ratio,

$$\mathbb{E}_{\pi_{\text{off}}}[\rho_\theta] = \sum_a \pi_{\text{off}}(a|s)\frac{\pi_\theta(a|s)}{\pi_{\text{off}}(a|s)} = \sum_a \pi_\theta(a|s) = 1, \tag{15}$$

which implies that the first-order term vanishes: $\mathbb{E}_{\pi_{\text{off}}}[\rho_\theta - 1] = 0$. Therefore, the leading term of the divergence is given by

$$\mathbb{E}_{\pi_{\text{off}}}[D_f(\pi_\theta \| \pi_{\text{off}})] = \frac{f''(1)}{2}\mathbb{E}_{\pi_{\text{off}}}[(\rho_\theta - 1)^2] + \mathcal{O}\left(\mathbb{E}_{\pi_{\text{off}}}[|\rho_\theta - 1|^3]\right). \tag{16}$$

Finally, since $\mathbb{E}_{\pi_{\text{off}}}[\rho_\theta] = 1$, the second central moment equals the variance:

$$\mathbb{E}_{\pi_{\text{off}}}[(\rho_\theta - 1)^2] = \text{Var}_{\pi_{\text{off}}}[\rho_\theta], \tag{17}$$

which completes the proof. $\square$

**Theorem A.2** (Variance Bound on Clipping Error)**.** *Let $\mathcal{J}^{\text{UNC}}(\theta) = \mathbb{E}_{(s_t,a_t)\sim\pi_{\text{off}}}[\rho_t(\theta)\hat{A}_t]$ be the unclipped off-policy objective, and $\mathcal{J}^{\text{CLIP}}(\theta)$ the clipped objective with clip range $\epsilon$. Assume that the support of $\pi_\theta$ is contained in that of $\pi_{\text{off}}$, and that $\mathbb{E}_{\pi_{\text{off}}}[\rho_t^2] < \infty$. Then,*

$$\left|\mathcal{J}^{\text{UNC}}(\theta) - \mathcal{J}^{\text{CLIP}}(\theta)\right| \leq \frac{A_{\max}}{\epsilon}\text{Var}_{\pi_{\text{off}}}[\rho_t(\theta)]. \tag{18}$$

*Proof.* We first observe that the unclipped and clipped objectives differ only when the policy ratio leaves the trust region $[1 - \epsilon, 1 + \epsilon]$. Define the absolute deviation

$$\Delta = \left|\mathcal{J}^{\text{UNC}}(\theta) - \mathcal{J}^{\text{CLIP}}(\theta)\right|.$$

By linearity of expectation and the triangle inequality,

$$\Delta \leq \mathbb{E}_{\pi_{\text{off}}}\left[\left|\rho_t(\theta)\hat{A}_t - \text{clip}(\rho_t(\theta), 1 - \epsilon, 1 + \epsilon)\hat{A}_t\right|\right]. \tag{19}$$

Using the boundedness of advantages $|\hat{A}_t| \leq A_{\max}$, we obtain

$$\Delta \leq A_{\max}\mathbb{E}_{\pi_{\text{off}}}\left[\left|\rho_t(\theta) - \text{clip}(\rho_t(\theta), 1 - \epsilon, 1 + \epsilon)\right|\right]. \tag{20}$$

The clipping residual can be written as

$$\left|\rho_t - \text{clip}(\rho_t, 1 - \epsilon, 1 + \epsilon)\right| = \max(0, \ |\rho_t - 1| - \epsilon).$$

For all $x$ such that $|x| > \epsilon$, we have

$$|x| - \epsilon \ \leq \ |x| = \frac{x^2}{|x|} \ \leq \ \frac{x^2}{\epsilon}, \tag{21}$$

where the last inequality follows from $|x| > \epsilon$. Applying this with $x = \rho_t - 1$ yields

$$\max(0, |\rho_t - 1| - \epsilon) \ \leq \ \frac{(\rho_t - 1)^2}{\epsilon}. \tag{22}$$

Substituting this bound back into the expectation gives

$$\Delta \leq \frac{A_{\max}}{\epsilon} \mathbb{E}_{\pi_{\text{off}}} \left[ (\rho_t(\theta) - 1)^2 \right] = \frac{A_{\max}}{\epsilon} \text{Var}_{\pi_{\text{off}}} [\rho_t(\theta)], \tag{23}$$

which completes the proof. □

## B. Variance Approximation of divergence

To theoretically justify replacing standard KL-divergence constraints with our variance-based regularization, we empirically analyze the relationship between the variance of the importance ratio $\rho_t(\theta) = \pi_\theta(a_t|s_t)/\pi_{\text{off}}(a_t|s_t)$ and various statistical divergences. As illustrated in Figure 5, we compare $\text{Var}(\rho_\theta)$ against six standard divergence metrics: Forward/Reverse KL-Divergence, Jensen-Shannon Divergence, Hellinger Distance, $\chi^2$-Divergence, and $\alpha$-Divergence ($\alpha = 0.5$). The plots reveal a strong linear correlation in the near-policy region, aligning with the second-order Taylor expansion where $D_f(\pi_\theta || \pi_{\text{off}}) \approx \frac{f''(1)}{2} \text{Var}(\rho_\theta)$. This confirms that penalizing the variance of $\rho_t$ effectively acts as a stable, computationally efficient trust-region constraint.

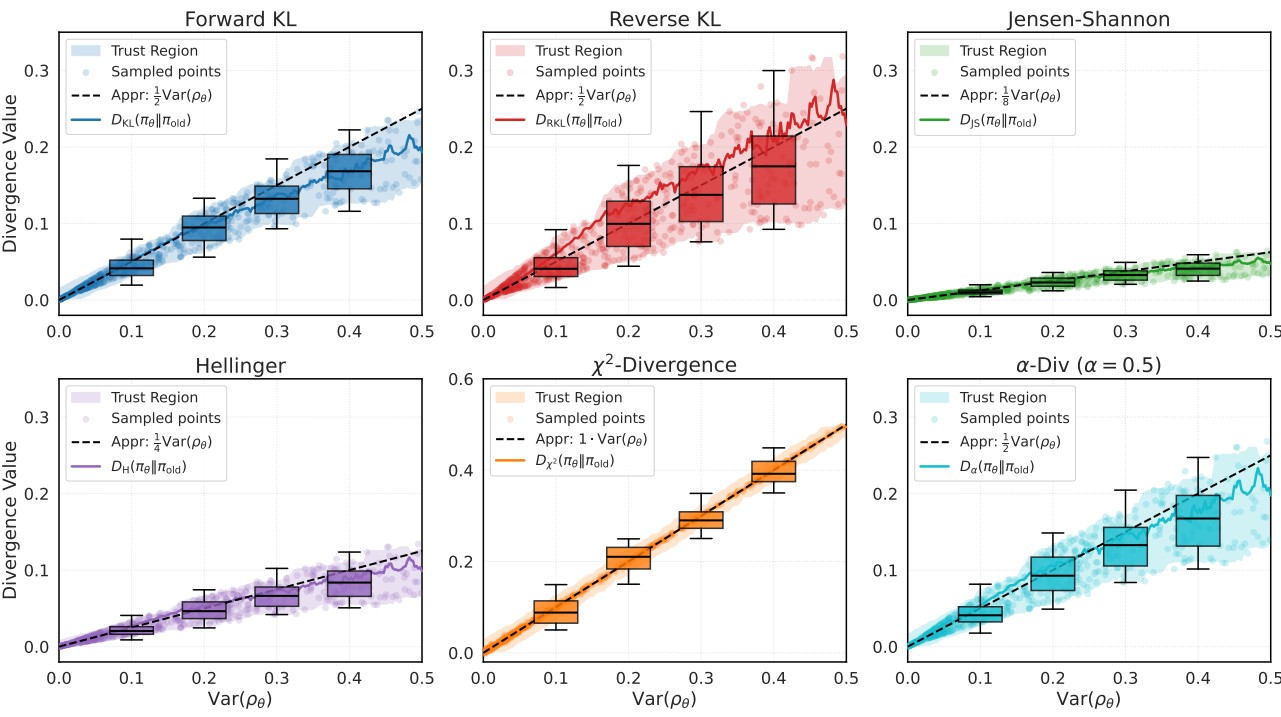

*Figure 5.* **Empirical Analysis of Variance as a Proxy for Divergence.** We visualize the relationship between the variance of policy ratios, $\text{Var}(\rho_\theta)$, and six common divergence metrics across sampled policy updates. The dashed black line represents the theoretical second-order approximation.

# C. Hyperparameters

We list the detailed hyperparameters used for both LLM mathematical reasoning and continuous robotic control experiments in Table 3.

*Table 3.* **Unified Hyperparameters.** Detailed configuration for LLM mathematical reasoning tasks (Part I) and MuJoCo Playground continuous control tasks (Part II). $R^2$VPO utilizes adaptive dual updates for LLMs and fixed dual factors for robotics tasks.

| Parameter | Value |
|---|:---:|
| **Part I: LLM Mathematical Reasoning** | |
| *Common Training Configuration* | |
| Optimizer | AdamW |
| Learning Rate | $1 \times 10^{-6}$ |
| Global Batch Size | 128 |
| Group Sampling Size ($G$) | 8 |
| Advantage Estimator | Relative Group |
| Max Model Length | 16384 |
| Gradient Accumulation Steps | 2 |
| Clip Gradient Norm | 1.0 |
| Training Temperature | 1.0 |
| Training Top-p | 1.0 |
| Training Top-k | -1 |
| *Test / Inference Configuration* | |
| Temperature | 0.6 |
| Top-p | 0.95 |
| Top-k | 20 |
| *$R^2$VPO Specifics (LLM)* | |
| Dual Factor Strategy | Adaptive (Primal-Dual) |
| Initial $\lambda$ | 0.0 |
| Dual Learning Rate ($\eta_\lambda$) | $5 \times 10^{-3}$ |
| Target Tolerance ($\delta$) | $1 \times 10^{-3}$ |
| *Off-Policy Settings:* | |
|     Replay Buffer Type | FIFO |
|     Replay Buffer Capacity | 4 Iterations |
|     Update-to-Data (UTD) Ratio | 2 |
|     Sampling Strategy | Uniform |
| **Part II: Continuous Robotic Control (DM Control)** | |
| *Training Configuration* | |
| Optimizer | Adam |
| Learning Rate | $1 \times 10^{-3}$ |
| Max Grad Norm | 2.0 |
| Episode Length | 1000 |
| Action Repeat | 1 |
| Rollout Length | 30 |
| Parallel Envs | 2048 |
| Minibatches | 32 |
| Epochs per Update | 16 |
| Batch Size | 1024 |
| *Algorithm Specifics* | |
| Discount Factor ($\gamma$) | 0.995 |
| Reward Scaling | 10.0 |
| Observation Normalization | True |
| Dual Factor ($\lambda$) | 0.06 (Fixed) |
| Network Architecture | MLP: (256)×5 (Policy) / (256)×5 (Value) |

# D. More Experimental Results

## D.1. Results on LLMs Training

We present the performance comparison of all LLMs across different benchmarks, as shown in Table 4.

*Table 4.* **Comprehensive Benchmarking Results.** We compare $R^2$VPO against multiple strong baselines (GRPO, GRPO-CH, GPPO, TOPR) across seven model scales. **Avg** reports the average accuracy, with the relative improvement over *Base* shown in parentheses.

| Method | AIME 24 | AIME 25 | AMC 23 | HMMT | OlymMath | Avg *(Gain vs Base)* |
|---|---|---|---|---|---|---|
| *openPangu-Embedded-1B* | | | | | | |
| Base | 20.83 | 21.67 | 60.00 | 9.59 | 4.00 | 23.22 |
| GRPO | 27.50 | 24.58 | 65.83 | 11.25 | 6.00 | 27.03 (+16.4%) |
| GRPO-CH | 28.75 | 25.83 | 67.08 | 11.43 | 7.00 | 28.02 (+20.7%) |
| GPPO | 30.41 | 25.42 | 62.92 | **11.67** | 5.50 | 27.18 (+17.1%) |
| TOPR | 26.66 | 25.00 | 63.33 | 10.84 | **7.50** | 26.67 (+14.9%) |
| $R^2$VPO-ON | **31.67** | 24.17 | 65.83 | 11.25 | 6.00 | 27.78 (+19.7%) |
| **$R^2$VPO-OFF** | 28.75 | **28.33** | **67.50** | 11.25 | 5.00 | **28.17** (+21.3%) |
| *Qwen3-1.7B-Fast* | | | | | | |
| Base | 14.17 | 12.50 | 45.00 | 4.58 | 3.50 | 15.95 |
| GRPO | 34.17 | 27.08 | 67.92 | 8.50 | 15.83 | 30.70 (+92.5%) |
| GRPO-CH | 36.67 | 31.25 | 73.33 | 11.00 | 16.67 | 33.78 (+111.8%) |
| GPPO | 39.58 | **35.42** | 77.92 | 10.50 | 18.75 | 36.43 (+128.4%) |
| TOPR | 36.25 | 33.75 | 71.25 | 13.00 | 19.17 | 34.68 (+117.4%) |
| $R^2$VPO-ON | 40.42 | **35.42** | **82.08** | 11.00 | 17.92 | 37.37 (+134.3%) |
| **$R^2$VPO-OFF** | **45.00** | 30.83 | 80.42 | **13.50** | **20.42** | **38.03** (+138.4%) |
| *DeepSeek-Distill-Qwen-1.5B* | | | | | | |
| Base | 29.58 | 23.75 | 72.50 | **13.75** | **8.00** | 29.52 |
| GRPO | 29.58 | 25.00 | 74.58 | 10.00 | 7.00 | 29.23 (-1.0%) |
| GRPO-CH | 32.08 | 25.42 | 77.92 | 12.92 | 6.00 | 30.87 (+4.6%) |
| GPPO | 33.33 | 26.67 | 77.50 | 11.25 | 6.50 | 31.05 (+5.2%) |
| TOPR | 35.83 | 25.83 | 75.83 | 12.08 | 6.50 | 31.21 (+5.7%) |
| $R^2$VPO-ON | 29.58 | 27.08 | 75.00 | 9.58 | 6.00 | 29.45 (-0.2%) |
| **$R^2$VPO-OFF** | **40.42** | **28.75** | **82.08** | 12.08 | 7.50 | **34.17** (+15.8%) |
| *openPangu-Embedded-7B* | | | | | | |
| Base | 51.67 | 42.92 | 76.25 | 24.17 | 13.00 | 41.60 |
| GRPO | 64.58 | 59.17 | 92.92 | 31.25 | 27.00 | 54.98 (+32.2%) |
| GRPO-CH | 67.92 | **61.25** | **93.75** | 32.08 | **32.50** | 57.50 (+38.2%) |
| GPPO | **70.42** | 60.83 | 91.88 | 28.33 | 29.00 | 56.09 (+34.8%) |
| TOPR | 68.33 | 55.00 | 92.81 | 33.33 | 28.50 | 55.60 (+33.6%) |
| **$R^2$VPO-ON** | 67.98 | 59.33 | 92.92 | **35.00** | **32.50** | **57.55** (+38.3%) |
| $R^2$VPO-OFF | 70.00 | 54.58 | 91.67 | 33.33 | 32.00 | 56.32 (+35.4%) |
| *Qwen3-8B-Fast* | | | | | | |
| Base | 25.83 | 21.67 | 68.75 | 12.92 | 7.50 | 27.33 |
| GRPO | 68.33 | 52.50 | 89.58 | 21.00 | 31.25 | 52.53 (+92.2%) |
| GRPO-CH | 68.33 | 57.08 | **95.42** | **27.50** | 27.08 | 55.08 (+101.5%) |
| GPPO | **71.25** | 59.17 | 95.00 | 22.50 | 30.83 | 55.75 (+104.0%) |
| TOPR | 69.58 | 55.42 | 94.58 | 26.00 | 29.58 | 55.03 (+101.3%) |
| $R^2$VPO-ON | 68.75 | 56.67 | **95.42** | 25.50 | 23.33 | 53.93 (+97.3%) |
| **$R^2$VPO-OFF** | 70.00 | **61.67** | **95.42** | 27.00 | **32.92** | **57.40** (+110.0%) |
| *Qwen3-4B-Thinking* | | | | | | |
| Base | 62.08 | 48.33 | 90.00 | **29.58** | **28.00** | 51.60 |
| GRPO | 67.08 | 54.58 | 90.83 | 21.67 | 16.50 | 50.13 (-2.8%) |
| GRPO-CH | 67.08 | 56.67 | 94.58 | 22.92 | 19.50 | 52.15 (+1.1%) |
| GPPO | 67.08 | 57.50 | **95.83** | 22.08 | 17.50 | 52.00 (+0.8%) |
| **TOPR** | **68.75** | **61.67** | 95.42 | 20.83 | 19.00 | **53.13** (+3.0%) |
| $R^2$VPO-ON | 68.33 | 60.00 | 95.00 | 23.33 | 18.00 | 52.93 (+2.6%) |
| $R^2$VPO-OFF | 63.75 | 56.25 | 94.58 | 25.42 | 20.00 | 52.00 (+0.8%) |
| *Qwen3-8B-Thinking* | | | | | | |
| Base | 60.83 | 46.67 | 87.50 | 32.08 | 29.00 | 51.22 |
| GRPO | 67.08 | 53.33 | 92.08 | 29.17 | 23.00 | 52.93 (+3.4%) |
| GRPO-CH | 72.08 | 59.17 | 92.92 | 33.33 | 26.00 | 56.70 (+10.7%) |
| GPPO | 70.00 | **62.50** | 93.75 | 23.33 | 24.00 | 54.72 (+6.8%) |
| TOPR | 70.00 | 57.50 | 91.67 | 34.17 | 25.00 | 55.67 (+8.7%) |
| $R^2$VPO-ON | **75.00** | **65.00** | **95.00** | 22.50 | 19.00 | 55.30 (+8.0%) |
| **$R^2$VPO-OFF** | 70.00 | 60.83 | 94.17 | **35.00** | **31.00** | **58.20** (+13.6%) |

To further investigate the optimization efficiency and behavioral changes induced by our method, we visualize the training dynamics across all seven base models.

**Comparison with an Exact KL Penalty.**    We additionally compare $R^2$VPO with an exact KL-penalty variant, which replaces the quadratic ratio-variance term with an empirical KL penalty. As shown in Figure 6, the two methods exhibit similar training dynamics in the studied regime, consistent with the observation that the empirical ratio distributions remain concentrated near one. This supports our interpretation that, in practical PPO/GRPO-style updates, the main trade-off is not improved global exactness, but simplicity and deployability: exact KL penalties preserve the full divergence geometry, whereas $R^2$VPO keeps the shared local second-order structure and implements it as a lightweight ratio-space regularizer.

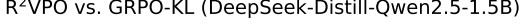

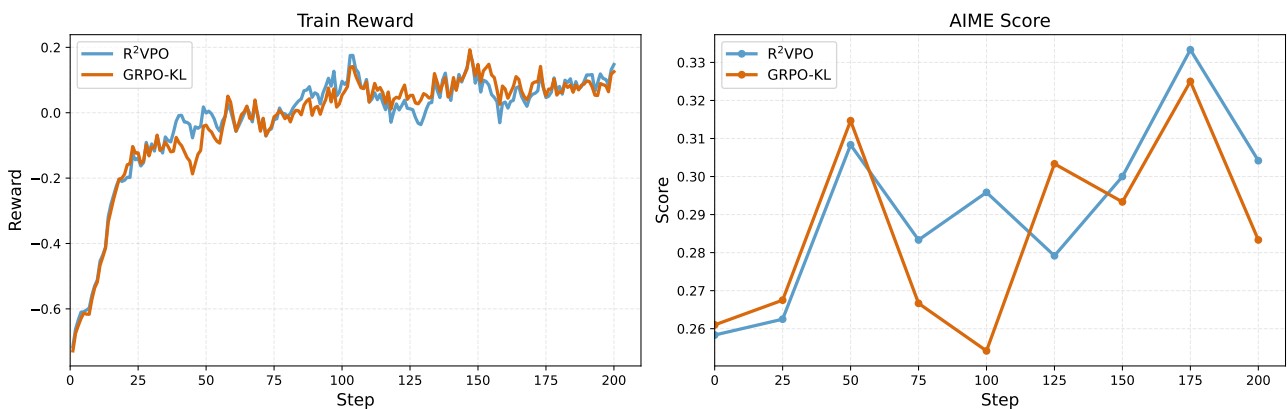

*Figure 6.* **Comparison with Exact KL Penalty.** We compare $R^2$VPO with a GRPO-KL variant on DeepSeek-Distill-Qwen2.5-1.5B. $R^2$VPO exhibits similar reward and AIME-score dynamics to the exact KL-penalty baseline, supporting the effectiveness of ratio variance as a lightweight local trust-region surrogate.

**Reward Convergence.**    As illustrated in Figure 7, $R^2$VPO exhibits superior sample efficiency. In both Fast and Slow thinking paradigms, $R^2$VPO-OFF consistently achieves faster reward convergence and higher final returns. This advantage is particularly pronounced in the *Qwen3-1.7B-Fast* and *DeepSeek-Distill-Qwen-1.5B* models, where baselines like GRPO struggle to improve or plateau early. This supports our claim that variance-aware regularization provides a robust signal for policy improvement, preventing the vanishing gradients often caused by aggressive clipping.

**Reasoning Length Evolution.**    Figure 8 tracks the average number of tokens per response. We observe two distinct behavioral shifts correlated with performance gains:

- **Expansion in Fast Thinking:** For models like *Qwen3-1.7B-Fast* and *Qwen3-8B-Fast*, $R^2$VPO triggers a rapid expansion in reasoning length. This aligns with their dramatic performance jumps, suggesting that the method successfully unlocks the models' capacity for extended reasoning chains.

- **Retention in Slow Thinking:** For models initialized with long-context capabilities (e.g., *Qwen3-4B-Thinking*), RL training naturally condenses the output. However, unlike GRPO, which often suffers from excessive length reduction (length collapse), $R^2$VPO preserves significantly more reasoning tokens. This indicates that our variance constraint protects valuable, complex reasoning paths from being prematurely pruned, maintaining the depth of thought required for hard benchmarks.

**Detailed Analysis of Ratio Dynamics.**    To further validate the stability of $R^2$VPO under off-policy conditions, we visualize the evolution of policy ratio distributions across varying degrees of data staleness ($rb \in \{0, 2, 4, 8\}$) and training phases in Figure 9. Crucially, the distributions remain tightly concentrated around $\rho_t = 1$ (indicated by the red vertical line) even at maximum staleness, providing strong empirical evidence that the optimization trajectory stays within the valid regime of our second-order variance approximation throughout the training process.

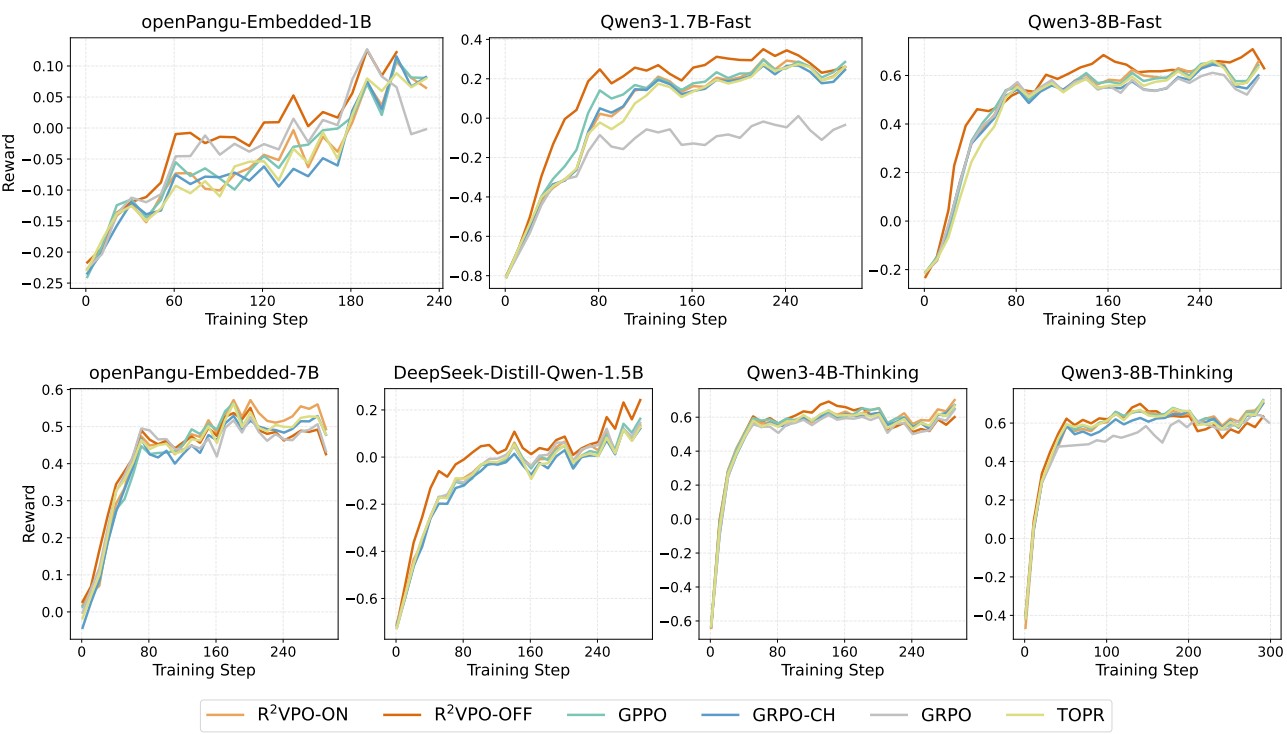

*Figure 7.* **Training Reward Dynamics across Seven Model Scales.** We compare the average episode reward curves of R²VPO (orange/brown) against baselines throughout the training process. R²VPO-OFF consistently demonstrates faster reward convergence and higher asymptotic performance, particularly on smaller models and distilled models, validating the efficiency of variance-regularized off-policy learning.

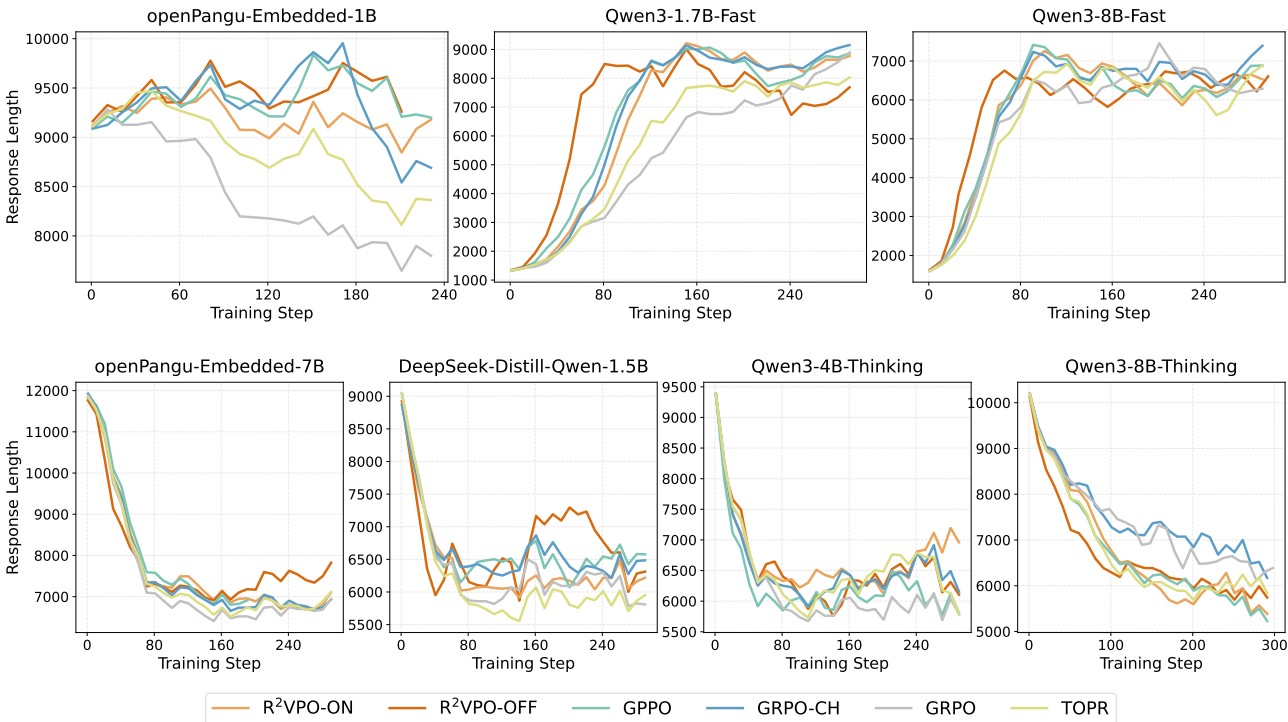

*Figure 8.* **Evolution of Response Length during Training.** The dynamics exhibit two distinct patterns: (1) For *Fast Thinking* models (top row), $R^2$VPO drives a substantial increase in response length, unlocking extended reasoning capabilities. (2) For *Slow Thinking* models (bottom row), while all methods reduce initial redundancy, $R^2$VPO maintains significantly longer reasoning chains than GRPO (grey line), which often suffers from length collapse. This suggests that variance-aware optimization effectively preserves complex reasoning paths that are otherwise truncated by heuristic clipping.

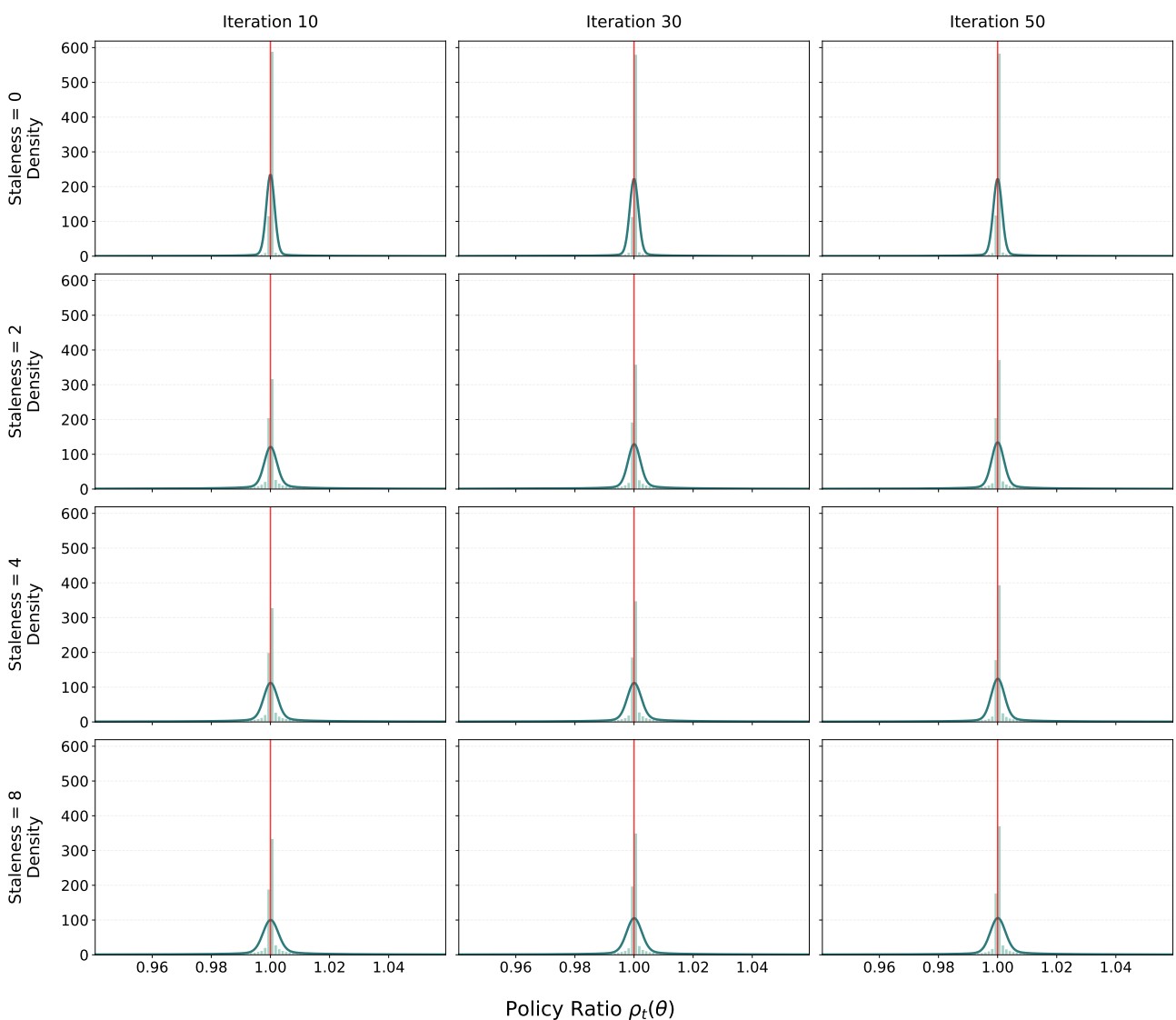

*Figure 9.* **Evolution of Policy Ratio Distributions.** The grid visualizes the density of $\rho_t(\theta)$ across training iterations (columns) and data staleness levels (rows). The consistent concentration around the center (red line at $\rho = 1$) confirms that $R^2$VPO effectively constrains divergence and prevents distributional collapse, even when reusing stale data from a large replay buffer ($rb = 8$).

## D.2. Results on Robotics Tasks

To provide a qualitative overview of our experimental settings, we visualize the scenes of the selected DeepMind Control Suite tasks in Figure 10. These environments cover a diverse range of challenges, including high-dimensional locomotion and sparse-reward manipulation.

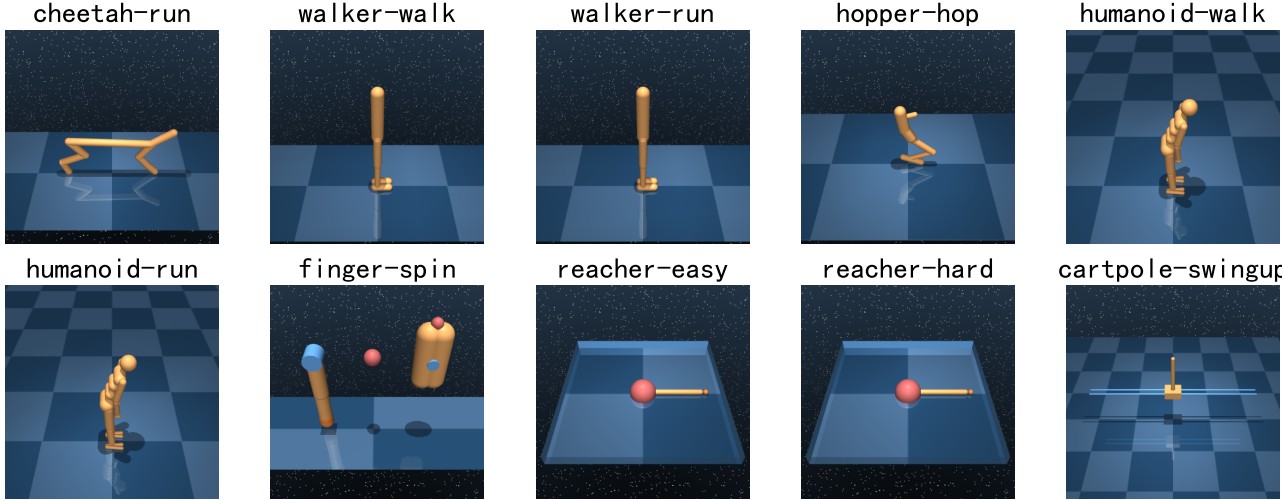

*Figure 10.* **Visualizations of the continuous control benchmarks.** We evaluate our method on 10 diverse tasks from the DeepMind Control Suite, ranging from simple balancing tasks (e.g., Cartpole) to complex locomotion tasks (e.g., Humanoid, Cheetah).

The full performance comparisons on these robotic tasks are reported in Figure 11. These results demonstrate that R$^2$VPO significantly outperforms PPO across the majority of tasks. In particular, R$^2$VPO exhibits superior sample efficiency and asymptotic performance in complex environments such as `WalkerRun` and `CheetahRun`, while maintaining robust learning in sparse-reward settings like `CartpoleSwingupSparse`.

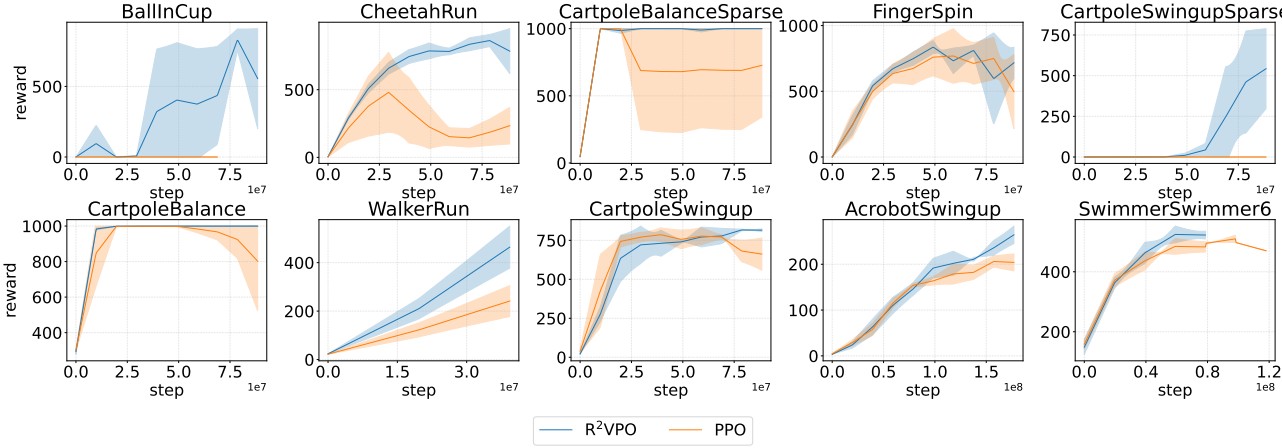

*Figure 11.* **Learning curves on DeepMind Control Suite tasks.** The x-axis denotes the number of environment steps, and the y-axis represents the average episode reward. Solid lines denote the mean performance over 5 independent training runs with different random seeds, and shaded regions denote standard deviation. R$^2$VPO (ours) consistently achieves higher returns and faster convergence compared to PPO.

**High-Dimensional Control with Stabilized Actor-Critic Training.** We further evaluate Humanoid-Run and Dog-Run in Figure 12, two more challenging high-dimensional continuous-control tasks. In our initial implementation, both vanilla PPO and vanilla R$^2$VPO fail to train reliably, suggesting that these tasks require additional actor-critic stabilization beyond the lightweight setup used in the main experiments. We therefore adopt a WPO (Pfau et al., 2025)-inspired stabilization setup.

Under this stronger configuration, both methods become trainable, and R$^2$VPO continues to outperform the corresponding PPO baseline. We note that Dog-Run remains sensitive to task-specific hyperparameters: the later-stage performance drop is likely caused by a conservative configuration that limits exploration after early improvement. A full environment-specific sweep may further improve both methods.

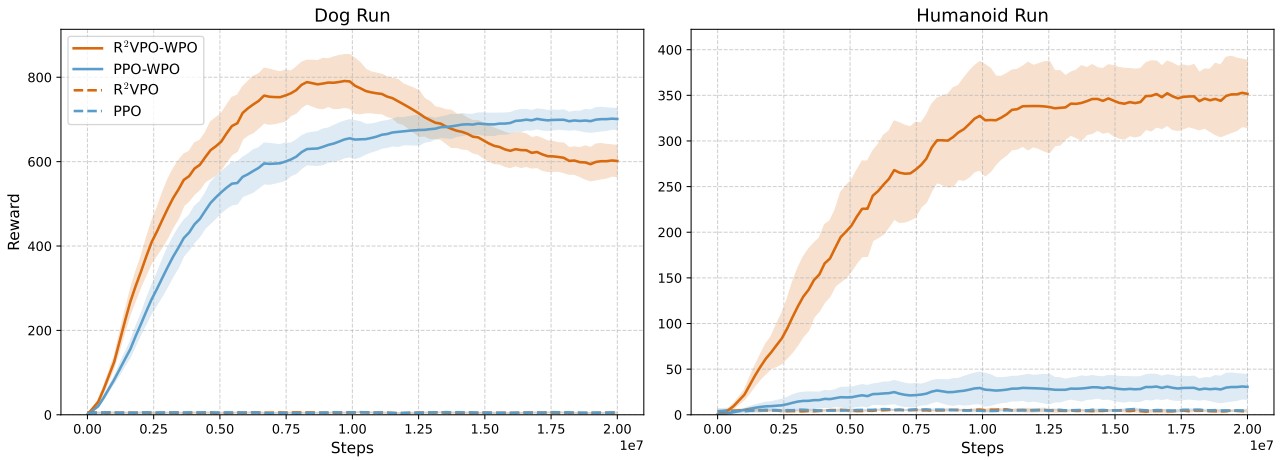

*Figure 12.* **High-Dimensional Control with Stabilized Training.** We compare R$^2$VPO and PPO on Dog-Run and Humanoid-Run under a WPO-inspired stabilization setup. R$^2$VPO achieves stronger performance than PPO on both tasks, while the later-stage decrease on Dog-Run suggests that this environment is sensitive to task-specific stabilization and regularization.

# E. Computational Infrastructure

All LLM fine-tuning experiments were conducted on a high-performance computing cluster consisting of **Huawei Ascend Atlas A3 nodes**.

To assess the computational overhead of our method, we report the average wall-clock time per training iteration across different model scales in Table 5. For the off-policy variant (**R$^2$VPO-OFF**), the inclusion of a replay buffer and a higher Update-to-Data (UTD) ratio of 2 results in slightly longer iteration times: approximately 300–400 seconds for 1B models, ≈500 seconds for 4B models, and 600–700 seconds for 7B/8B models.

In contrast, the on-policy variant (**R$^2$VPO-ON**) requires approximately 50–100 seconds less per iteration than its off-policy counterpart. Crucially, the runtime of R$^2$VPO-ON is strictly comparable to other baseline methods (e.g., GRPO), confirming that the calculation of the variance penalty introduces negligible computational overhead.

*Table 5.* **Average Training Runtime per Training Step.** Comparison of wall-clock times across different model scales on 2 Huawei Ascend Atlas A3 nodes. R$^2$VPO-ON maintains parity with standard baselines, while R$^2$VPO-OFF incurs a modest increase due to multiple updates per data batch (UTD=2).

| Model Scale | Method | Time per Iteration (s) |
|---|---|---|
| 1B Scale (e.g., Pangu-1B, Qwen3-1.7B) | R$^2$VPO-ON 
 R$^2$VPO-OFF | ≈ 250 − 300 
 300 − 400 |
| 4B Scale (e.g., Qwen-4B) | R$^2$VPO-ON 
 R$^2$VPO-OFF | ≈ 400 − 450 
 ≈ 500 |
| 7B/8B Scale (e.g., Pangu-7B, Qwen-8B) | R$^2$VPO-ON 
 R$^2$VPO-OFF | ≈ 550 − 600 
 600 − 700 |

