# OpenReview forum: "Ratio-Variance Regularized Policy Optimization"
_ICML.cc/2026/Conference — ICML 2026 spotlight_

### Official Review · Reviewer_nwUm · 2026-02-24

**Soundness:** 3
**Presentation:** 4
**Significance:** 3
**Originality:** 3
**Overall Recommendation:** 4
**Confidence:** 4

**Summary:**

This paper proposes R2VPO (Ratio-Variance Regularized Policy Optimization), a trust-region RL method that replaces PPO's heuristic clipping with variance regularization of the policy ratio \(\rho_t(\theta)\). It derives variance as a second-order approximation to f-divergences (Prop. 3.1), formulates a primal-dual objective (Thm. 3.3), and bounds PPO clipping error by variance (Thm. 3.4). R2VPO supports both on-policy and off-policy variants via soft penalties, evaluated on LLM reasoning (7 models, +35% macro gain) and robotic control (10 DM Control tasks).

**Compliance With Llm Reviewing Policy:**

Affirmed.

**Key Questions For Authors:**

1. Thm. 3.4 assumes \(\pi_\theta\) support \(\subset \pi_\text{off}\); how robust empirically when violated (e.g., LLM mode collapse)? Breakdown by ratio violation could validate/strengthen soundness.

2. No theory for global improvement/monotonicity; does R2VPO guarantee ascent like TRPO? Local approx. analysis would raise soundness to excellent.

3. Off-policy gains strongest on small LLMs; compute-normalized expts (FLOPs/rollouts) vs. GRPO on large models? Positive reply boosts significance.

4. Adaptive \(\lambda\) key (Fig. 4d); sensitivity to init \(\eta_\lambda\)/timescale? Ablation table could confirm robustness.

**Limitations:**

Yes—implicit via analysis (local approx., no global theory); standard impact statement (no societal issues).

**Strengths And Weaknesses:**

**Soundness**: Good. Prop. 3.1 correctly derives variance approx. under \(\rho_t \to 1\) (standard Taylor); Thm. 3.4 bound is novel but assumes bounded support/advantages (reasonable). Experiments comprehensive: 7 LLMs across paradigms, 10 robotics tasks, 5 seeds, ablations on staleness/adaptive \(\lambda\). Claims supported (e.g., Fig. 4 validates mechanism), but lacks monotonic improvement proof (unlike TRPO); off-policy gains rely on empirical staleness tolerance. Authors honest about heuristics in baselines.

**Presentation**: Excellent. Clear flow: approx. → dual → algos → expts. Intuitive figs (2-4); Table 1 unifies divergences; Alg. 1 explicit/reproducible. Related work (Sec. 5) positions vs. clipping variants (GRPO-CH, GPPO). Minor: hyperparameters in App. C could tabulate more crisply.

**Significance**: Good. Tackles PPO clipping flaws (signal loss, no off-policy reuse), relevant for LLM-RLHF/robotics scaling. +35% reasoning gains (esp. small models) + sample efficiency could influence practice; off-policy decoupling aids async training. Specialized to policy gradient RL but timely for 2026 RLHF trends.

**Originality**: Good. Insightful: variance as *unified* f-divergence proxy + dual for adaptive control. Distinguishes from patches (GPPO/TOPR) via theory (Thm. 3.4); off-policy via natural penalty (no critics). Builds on trust-region approx. but recombines with Lagrangian for generality.

---

> ### Author Rebuttal · Authors · 2026-03-30
>
> Thank you very much for your positive and detailed feedback. We especially appreciate your recognition of the theory, mechanism analysis, and practical relevance of R$^2$VPO.
>
> > **Q1: Thm. 3.4 assumes $\pi_\theta$ support $\subset \pi_{\mathrm{off}}$; how robust is the method when this is violated?**
>
> Thank you for the question. In our practical setting, this condition is usually naturally satisfied on the training samples: in both the on-policy and off-policy variants, the data are collected by the behavior policy $\pi_{\mathrm{off}}$, so for every sampled $(s,a)$ we have $\pi_{\mathrm{off}}(a\mid s)>0$. Hence the importance ratio $\rho=\pi_\theta/\pi_{\mathrm{off}}$ is well-defined, i.e., we do not encounter $\pi_\theta(a\mid s)>0$ while $\pi_{\mathrm{off}}(a\mid s)=0$.
>
> In practice, the more relevant issue is whether policy drift makes $\pi_{\mathrm{off}}(a\mid s)$ very small and induces extreme ratios / heavy-tailed weights. Our results suggest this does not become severe in the studied regime: Fig. 4(c) and Fig. 8 show that ratios remain bounded and concentrated near $\rho=1$ even with stale-data reuse. This is exactly where the variance penalty helps, by suppressing such extreme-ratio behaviors before they become destabilizing. We will clarify this in the revision.
>
> > **Q2: No theory for global improvement/monotonicity; does R$^2$VPO guarantee ascent like TRPO?**
>
> We do **not** claim a TRPO-style global monotonic improvement guarantee. TRPO relies on the exact trust-region constraint, whereas our method replaces it with a **local second-order surrogate** based on ratio variance and optimizes the resulting objective in a primal-dual form.
>
> What we do claim is a **local trust-region interpretation**: Prop. 3.1 shows that when $\rho\approx 1$, the exact $f$-divergence is well approximated by the quadratic ratio-variance term up to a higher-order remainder. In this sense, R$^2$VPO preserves the same local trust-region control logic as TRPO while trading exact global guarantees for a much simpler first-order surrogate in ratio space. This is also supported empirically by Fig. 2, Fig. 4(c), and Fig. 8. We will clarify this positioning in the revision.
>
> > **Q3: Off-policy gains are strongest on small LLMs; do compute-normalized experiments on larger models support the significance claim?**
>
> Yes. Following your suggestion, we additionally computed the cost required to reach matched performance thresholds, and compared off-policy R$^2$VPO against GRPO on **Pangu-7B** and **Qwen3-8B**:
>
> | Model    | Target    | Method       | FLOPs               | Tokens             | Steps |
> | -------- | --------- | ------------ | ------------------- | ------------------ | ----- |
> | Pangu-7B | $\sim 52$ | R$^2$VPO-OFF | 2.17E19 (**46.0%**) | 2.21E8 (**46.2%**) | 100   |
> |          |           | GRPO         | 4.02E19             | 4.11E8             | 150   |
> | Qwen3-8B | $\sim 56$ | R$^2$VPO-OFF | 5.80E19 (**42.0%**) | 9.06E8 (**42.3%**) | 125   |
> |          |           | GRPO         | 1.00E20             | 1.57E9             | 200   |
>
> Thus, the off-policy benefit is **not** merely a small-model artifact: even on larger 7B/8B settings, R$^2$VPO-OFF reaches comparable quality with substantially lower compute and fewer generated tokens. We will include the full compute-normalized comparison in the revision.
>
> > **Q4: How sensitive is adaptive $\lambda$ to $\eta_\lambda$ / timescale?**
>
> To check robustness, we ablated the dual learning rate $\eta_\lambda$ in the adaptive-$\lambda$ setting. Entries below are **AIME24 / Reward**:
>
> | Step | $5e{-3}$    | $1e{-3}$    |
> | ---- | ----------- | ----------- |
> | 25   | 0.52 / 0.46 | 0.44 / 0.38 |
> | 50   | 0.58 / 0.47 | 0.55 / 0.44 |
> | 75   | 0.59 / 0.56 | 0.54 / 0.47 |
> | 100  | 0.61 / 0.64 | 0.57 / 0.60 |
>
> Corresponding curves are shown in https://anonymous.4open.science/r/More_results_for_R2VPO-FF82/Figure4.md. Both settings are stable and improve throughout training, while $5e{-3}$ consistently performs better. This suggests that adaptive $\lambda$ is **not brittle** within the studied range, though a moderately larger dual step improves responsiveness and final performance. We will include this ablation in the revision.

---

> > ### Author Rebuttal · Reviewer_nwUm · 2026-04-03
> >
> > The writer addressed all the claims

---

> > > ### Author Response · Authors · 2026-04-04
> > >
> > > Dear reviewer,
> > >
> > > Thank you for your reply! We will improve our paper considering your comments accordingly. We sincerely appreciate you for your insightful comments.
> > >
> > > Best wishes!
> > >
> > > The authors

---

### Official Review · Reviewer_YRpk · 2026-02-26

**Soundness:** 3
**Presentation:** 2
**Significance:** 2
**Originality:** 3
**Overall Recommendation:** 4
**Confidence:** 3

**Summary:**

This work introduces Ratio-Variance Regularized Policy Optimization (R2VPO), which is an on-policy reinforcement learning method that employs local approximations to a trust-region constraint by considering the policy ratio variance rather than other methods, such as clipping likelihood ratios. The method is evaluated on various domains, such as LLM evaluations or robot control environments.

**Compliance With Llm Reviewing Policy:**

Affirmed.

**Final Justification:**

Based on the answers, I have adjusted my score from 3 to 4.

**Key Questions For Authors:**

- How do the bounds vary for the exact and the second-order approximations of the bounds? Is it necessary to be more conservative with the proposed method? A comparison to exact trust region methods such as [1] would reveal how much better the proposed method is in terms of conservativeness introduced by the approximation.

- How does the method behave in high-dimensional continuous control tasks? While I appreciate that the method was run on DMC's continuous control domains, I don't think the considered environments are representative of difficult, high-dimensional control environments. An interesting analysis would, for example, be the humanoid-run or the dog-run environments on the DMC suite.

- From Table 1, it becomes clear that the approximations for all divergences are locally the same up to a constant scaling factor. I am wondering how small the epsilon around the ratio rho=1 needs to be such that this approximation holds, given that the divergences in general behave differently globally, but essentially behave similarly in a small, local region. This question is also related to the conservativeness of the constraint, ensuring it works, as mentioned in the question before. I think it is necessary to compare this method against exact trust region methods [1] to answer this question, which, in my opinion, plays an important role in evaluating the paper's significance.

I am ready to increase my score if my questions and concerns are addressed.

**Limitations:**

The limitations and the societal impact are adequately addressed in the paper.

**Strengths And Weaknesses:**

Strengths:

The paper considers an interesting relation for f-divergence approximations that serve as a trust-region in policy optimization. It is a relevant research field to improve the performance and the data efficiency in reinforcement learning.


Weaknesses:

While I agree that PPO and its related clipping objective is a very common and widely used method, it is important to note that PPO is not the only well-performing trust-region method in RL. Hence, the paper needs a discussion of important related works. For example, in on-policy learning, trust region constraints have already been enforced in the relative entropy policy search framework and their follow-up works without any clipping:

	- Jan Peters et al., Relative entropy policy search, AAAI Conference on Artificial Intelligence 2010
	- Herke van Hoof et al., Non-parametric Policy Search with Limited Information Loss, JMLR 2017

A newer line of work even considers differentiable trust-region layers for deep neural networks and also discusses the benefits over the PPO clipping objective, where different constraint types are also discussed:

	- Fabian Otto et al., Differentiable Trust Region Layers for Deep Reinforcement Learning. ICLR 2021

Trust region constraints have also been discussed in the Maximum A Posteriori framework in the on-policy setting:

	- H. Francis Song et al., V-MPO: On-Policy Maximum A Posterior Policy Optimization for Discrete and Continuous Control.

As a reader, I believe it would also be important to discuss the exact relation to natural gradients in RL, as these methods are based on the second-order approximation of the trust region as considered here. I think a clear discussion of the differences beyond using f-divergences is important. Especially, given that the dual formulation of the constrained optimization problem was also already considered in the context of trust-region policy optimization, for e.g. Peters et al. (see above references).

---

> ### Author Rebuttal · Authors · 2026-03-30
>
> We sincerely thank the reviewer for the thoughtful comments. We agree that the paper should better position itself within the broader trust-region literature, and we appreciate the opportunity to clarify both the novelty and the scope of our method.
>
> > W1: Relation to REPS / natural gradient / trust-region layers / V-MPO and broader trust-region literature
>
> Thank you for this important suggestion. We agree that PPO is not the only strong trust-region method, and that the current version should more clearly discuss prior non-clipping approaches.
>
> We do **not** claim to be the first non-clipping trust-region method, nor the first dual constrained policy optimization method. Earlier methods such as REPS already optimize under explicit KL / information-loss constraints; natural-gradient / TRPO-style methods are rooted in the local second-order geometry of KL trust regions; differentiable trust-region layers and V-MPO are also important related approaches.
>
> Our contribution is instead a **ratio-based instantiation** of this idea: for a broad class of $f$-divergences, we show that the shared local trust-region geometry is captured by the variance of the policy ratio, and we turn this into a simple clip-free quadratic regularizer **directly in importance-ratio space**. Unlike natural-gradient / TRPO-style methods, which realize trust regions through Fisher-preconditioned or explicitly constrained parameter updates, our method operates on the ratio already present in PPO/GRPO-style objectives. This makes it easy to integrate into modern RL pipelines, especially LLM post-training and replay-based training, without second-order solvers, explicit projection layers, or Fisher-matrix estimation. We will revise the Introduction and Related Work accordingly.
>
> > Q1: Exact vs. second-order trust-region approximation and conservativeness
>
> Thank you for this important question. Our method is intended as a **local second-order surrogate**, rather than an exact trust-region update. As stated in Prop. 3.1, the exact $f$-divergence and our quadratic surrogate agree up to second order around $\rho=1$, with approximation gap $O(\mathbb{E}[|\rho-1|^3])$. Thus, the surrogate is accurate when ratios stay close to 1, and becomes less exact only when updates move farther away.
>
> Accordingly, we do **not** claim the global exactness of methods such as REPS. The trade-off is that exact methods preserve the full divergence geometry, while our method keeps only its local second-order structure and turns it into a tractable quadratic regularizer in ratio space. Empirically, Fig. 2 and Fig. 5 show close agreement between exact divergences and their quadratic surrogates, while Fig. 4(c) and Fig. 8 show that observed ratio distributions remain tightly concentrated near 1 during training, including under stale-data / off-policy reuse. We provide a comparison against an exact KL-based penalty baseline in https://anonymous.4open.science/r/More_results_for_R2VPO-FF82/Figure2.md; the behavior is very similar, supporting our point that the practical trade-off is mainly exactness vs. simplicity/deployability.
>
> > Q2: Behavior in higher-dimensional continuous control tasks
>
> Thank you for this important suggestion. We evaluated the Humanoid-Run and Dog-Run tasks. In our runs, both the original R$^2$VPO and the original PPO baseline fail to train reliably on these two environments, suggesting that they require stronger stabilization than the setup used in the main paper. We therefore further adopted an **actor-critic/value-based stabilization setup inspired by WPO [5]**. Under this stronger setup, training becomes effective, and R$^2$VPO continues to outperform the corresponding PPO baseline on both tasks (Please refer to https://anonymous.4open.science/r/More_results_for_R2VPO-FF82/Figure3.md). We will include these additional results and discussion in the revision.
>
> [5] Wasserstein Policy Optimization.
>
> > Q3: Local validity of the $\rho \approx 1$ approximation and its significance
>
> Thank you for this question. Table 1 indeed shows that a broad class of $f$-divergences share the same local second-order structure around $\rho=1$, up to the scaling factor $\frac{f''(1)}{2}$. This is exactly the motivation of our method: although these divergences differ globally, they share the same **local trust-region geometry**, which can be captured by ratio variance.
>
> In our view, the significance of the method does **not** come from being more globally exact than exact trust-region methods such as REPS. Rather, it comes from turning this shared local structure into a simple clip-free regularizer directly in importance-ratio space. This is particularly useful for modern ratio-based RL—especially LLM post-training—because the regularized quantity is exactly the ratio already present in PPO/GRPO-style objectives. As a result, the method can be integrated into existing training pipelines while naturally supporting replay / stale-data reuse.

---

> > ### Author Rebuttal · Reviewer_YRpk · 2026-04-01
> >
> > I thank the authors for their response and their efforts in additional results. I am adjusting my score, and I would be happy to see that the paper discusses the aforementioned related works/exact trust region methods. An interesting observation is that the orange curve for the dog run environment seems to decrease in performance over the number of samples. Do the authors have an intuition why this happens?

---

> > > ### Author Response · Authors · 2026-04-02
> > >
> > > Dear reviewer,
> > >
> > > Thank you again for your thoughtful follow-up and for adjusting your score. We also appreciate your positive acknowledgement of our rebuttal.
> > >
> > > Our current intuition is that Dog-Run is quite sensitive to the training configuration, and the setup used here may still be somewhat conservative, which can limit exploration after the initial improvement stage and lead to the later performance drop. This is also broadly consistent with WPO, where the authors perform a hyperparameter sweep to obtain settings that work well across control-suite tasks, and note that environment-specific tuning can further improve performance. In our case, we adopted a WPO-inspired actor-critic/value-based stabilization setup, but did not yet fully sweep the task-specific hyperparameters for Dog-Run / Humanoid-Run. We will add this discussion in the revision.
> > >
> > > Best wishes,
> > >
> > > The authors

---

### Official Review · Reviewer_bVJT · 2026-02-26

**Soundness:** 4
**Presentation:** 4
**Significance:** 4
**Originality:** 4
**Overall Recommendation:** 6
**Confidence:** 4

**Summary:**

Policy gradient RL methods can be paired with trust region protections to enhance stable and provide a monotonic learning function. The authors propose a trust region protection method that avoids expensive second-order optimization and the binary thresholding of first-order methods like PPO: constraining policy ratio variance. The paper demonstrates that the divergence is dominated by the variance of the policy ratio. The proposed objective, R2VPO, is the policy gradient term (policy ratio times advantage) with an added regularization term to penalize the ratio for diverging from 1.

**Compliance With Llm Reviewing Policy:**

Affirmed.

**Final Justification:**

The rebuttal addressed my concerns and I have raised my score as indicated in response. I have no further concerns with this paper.

**Key Questions For Authors:**

1. In line with Weakness 1, how stable and reliable is the algorithm across training runs?
2. Will the authors publish the experiment code once the paper is deanonymized?
3. Off-policy learning and dual descent to learn lambda improve end performance and increase the rate of improvement in terms of training steps, but how much of a computational overhead do dual descent and off-policy add? Does the model achieve high performance faster in wall-clock time without these improvements than with them?

**Limitations:**

Yes

**Strengths And Weaknesses:**

Strengths:
1.  Clearly demonstrates theoretical reasoning behind the algorithm
2. R2VPO is simpler than the alternative PPO, replacing hard clipping with a regularization term
3. Empirical results are clear and show consistent improvement in various domains, including the high-importance domains of continuous control and LLM fine-tuning
4. The mechanism analysis is particularly useful in proving the benefits of this method

Weaknesses:
1. In the continuous control experiments, standard deviation across validation seeds is provided. However, none of the experiments show deviation across training runs. Given how prone reinforcement learning methods like PPO are to failed training runs and optimization issues, it is important to demonstrate multiple training runs to prove the algorithm's reliability (though its success across various environments somewhat demonstrates this).

Minor issues:
1. In Table 1, Reverse KL and Forward KL’s labels are switched in the far right column (F_KL and R_KL respectively)
2. The Related Works section would be more useful after the introduction, not at the end.

---

> ### Author Rebuttal · Authors · 2026-03-30
>
> Thank you very much for your highly positive assessment and for the constructive suggestions. We especially appreciate your comments on reliability, reproducibility, and computational cost, and we agree that clarifying these practical aspects will further strengthen the paper.
>
> > W1 & Q1: Demonstrating multiple training runs to establish the algorithm’s reliability
>
> Thank you for this important suggestion. Our continuous-control figures already report variability across **multiple independent training runs**: the shaded regions in Fig. 3 / Fig. 10 denote the standard deviation over **5 random seeds**, rather than variation within a single run. We agree that this was not stated clearly enough, and we will revise the captions to explicitly say “mean ± std over 5 independent runs.”
>
> To further address your concern about run-to-run reliability, we additionally provide the individual training curves from these 5 runs for several representative tasks in https://anonymous.4open.science/r/More_results_for_R2VPO-FF82/Figure1.md.
>
> These per-run curves show that, compared with PPO, R$^2$VPO has more consistent convergence, lower failure/collapse frequency, and smaller final-performance variance. This is especially visible in the same regimes highlighted in the paper: sparse-reward tasks such as Ball-in-Cup and unstable settings such as Cheetah-Run, where PPO is more prone to failed or collapsed runs while R$^2$VPO remains consistently stable. We will make this point more explicit in the revision.
>
> > W2: In Table 1, Reverse KL and Forward KL’s labels are switched in the far-right column
>
> Thank you for catching this typo. We will correct the Forward/Reverse KL labels in Table 1 in the revised version.
>
> > W3: The Related Works section would be more useful after the introduction, not at the end
>
> Thank you for this suggestion. We agree that placing Related Work earlier would improve readability, and we will move it accordingly in the revised version.
>
> > Q2: Will the authors publish the experiment code once the paper is deanonymized?
>
> Yes. We plan to release the training code, main experiment configurations, and scripts used for evaluation once the paper is deanonymized.
>
> > Q3: What computational overhead do dual descent and off-policy add? Does the model achieve high performance faster in wall-clock time without these improvements than with them?
>
> Thank you for raising this practical question.
>
> For **dual descent**, the overhead is negligible. Once the policy ratio has been computed, the dual update only applies Eq. (10) to update a single scalar variable $\lambda$, which adds almost no measurable cost relative to the main forward/backward passes of policy optimization.
>
> For **off-policy learning**, the objective itself remains the same as Eq. (7); the additional cost mainly comes from replay-buffer storage/sampling and from the higher update-to-data ratio used in R$^2$VPO-OFF. As reported in Appendix E / Table 5, this results in only a modest increase in per-iteration wall-clock time:
>
> | Model scale | R$^2$VPO-ON (s/iter) | R$^2$VPO-OFF (s/iter) | Extra overhead of OFF |
> | ----------- | -------------------- | --------------------- | --------------------- |
> | 1B          | 250–300              | 300–400               | +50–150s              |
> | 4B          | 400–450              | $\approx$500          | +50–100s              |
> | 7B–8B       | 550–600              | 600–700               | +50–150s              |
>
> These numbers show that the overhead is modest rather than fundamental.
>
> At the same time, the training dynamics in Fig. 6 show that R$^2$VPO-OFF typically converges in **fewer optimization steps** and reaches higher final reward/performance, especially on smaller and distilled models. Therefore, although each iteration is somewhat slower, the improved sample efficiency substantially compensates for this cost. In the regimes where off-policy reuse is most beneficial, R$^2$VPO-OFF reaches a given performance level in comparable or shorter wall-clock time, while also achieving stronger final performance.
>
> We will clarify this trade-off more explicitly in the revision by jointly discussing the runtime table and the convergence curves.

---

> > ### Author Rebuttal · Reviewer_bVJT · 2026-03-31
> >
> > This clarifies all the issues I had with the paper. Assuming the additional results will be added to the paper or the appendix, I have increased the Soundness score in response.

---

> > > ### Author Response · Authors · 2026-04-01
> > >
> > > Dear reviewer,
> > >
> > > We deeply appreciate your thorough review and valuable suggestions, which are of great help to improve the quality of our work!
> > >
> > > Best wishes!
> > >
> > > The authors

---

### Official Review · Reviewer_t29D · 2026-03-13

**Soundness:** 3
**Presentation:** 3
**Significance:** 3
**Originality:** 3
**Overall Recommendation:** 5
**Confidence:** 2

**Summary:**

This paper introduces $R^2VPO$, a principled framework designed to overcome the fundamental limitations of heuristic clipping in standard reinforcement learning algorithms like PPO. The authors argue that binary hard clipping is essentially a "pointwise" decision that indiscriminately truncates high-return updates if they induce high policy divergence, effectively zeroing out critical gradient signals from novel, high-value discoveries.

To address this, the research theoretically establishes that the local geometry of various trust-region constraints (f-divergences) is approximately governed by the variance of the policy ratio. By leveraging this insight, the authors recast the optimization problem into a primal-dual framework using Lagrangian multipliers. This approach replaces the rigid clipping boundary with a "soft brake"—a quadratic penalty that adaptively scales updates based on their divergence rather than abruptly stopping them.

The resulting algorithm proves significantly more sample-efficient and stable. Extensive evaluations across seven LLM scales and ten robotic control tasks demonstrate that $R^2VPO$ successfully preserves informative signals that PPO would otherwise discard. As the authors have noted, this advantage is particularly pronounced in sparse-reward environments and complex reasoning tasks, where the ability to "softly" regularize rare but successful explorations allows the model to achieve superior asymptotic performance and robustness against data staleness.

**Compliance With Llm Reviewing Policy:**

Affirmed.

**Final Justification:**

The authors have provided a well-structured rebuttal.

The clarification on the dual variable λ — demonstrating robust performance across both fixed and adaptive variants — resolves my only noted weakness. The discussion on PPO's competitive regimes was also informative, confirming that R²VPO's advantage is most pronounced precisely when PPO's clipping becomes the optimization bottleneck, particularly in sparse-reward control and LLM training settings. The theoretical contribution of unifying f-divergence constraints through variance of the policy ratio remains strong, and the extensive empirical validation across both robotics and LLM scales is convincing.

Therefore, I update my score to Accept.

**Key Questions For Authors:**

**Q1**:It would be highly beneficial for future practitioners to understand the practical boundaries of the proposed approach.

- Could the authors discuss any specific scenarios or benchmarks where the standard PPO baseline demonstrated superior performance or more favorable characteristics (e.g., initial convergence speed or robustness to early-stage stochasticity) compared to $R^2VPO$? Such insights would provide a more balanced view of the algorithm's applicability.

**Limitations:**

Yes

**Strengths And Weaknesses:**

The proposed framework replaces the heuristic clipping of PPO with a principled variance-based constraint. The expansion of this RL algorithm into the LLM domain is particularly significant, as LLM reasoning tasks often represent the ultimate sparse reward setting, where a long chain of correct tokens is required to achieve a single successful outcome.

**Strength**
**Preservation of High-Value Exploration**: Unlike PPO's binary clipping, which indiscriminately zeros out gradients for updates exceeding a fixed threshold, $R^2VPO$ acts as a distributional "soft brake". This allows the model to preserve critical gradient signals from novel, high-return discoveries—such as a robot executing a complex maneuver or an LLM finding a unique reasoning path—while still preventing destructive policy shifts.

**Principled Theoretical Foundation**: The authors prove that the local geometry of a broad class of trust-region constraints ($f$-divergences) is universally governed by the variance of the policy ratio. This allows for a unified, computationally tractable approximation of various divergences (like KL or $\chi^2$) using a simple quadratic penalty.

**Superiority in Sparse and Dynamic Environments**: Empirical evaluations show that $R^2VPO$ significantly outperforms PPO in challenging sparse-reward tasks like Ball-in-Cup, where PPO often fails to learn entirely. In the LLM domain, it successfully unlocks extended reasoning capabilities in smaller models, achieving large gain in accuracy.

**Weakness**

While there is little to criticize, a minor point is that a proper policy update depends on the dual variable (lambda) being accurately updated beforehand, requiring the learning rate discrepancy between the two to be carefully considered. However, this is an inherent and acceptable trade-off for any Lagrangian-based optimization framework.

---

> ### Author Rebuttal · Authors · 2026-03-30
>
> We sincerely thank the reviewer for the positive assessment and thoughtful comments. We especially appreciate the reviewer’s recognition of the central motivation of our work: replacing pointwise hard clipping with a smoother, variance-aware trust-region surrogate that better preserves informative high-return updates.
>
> > W1: About the dual variable ($\lambda$) and training stability
>
> We thank the reviewer for this insightful point. We agree that handling the dual variable is an important practical consideration shared by Lagrangian-based methods.
>
> Empirically, however, our results suggest that R$^2$VPO is not brittle to this design choice. In the paper, we study both a fixed-$\lambda$ variant and an adaptive-$\lambda$ variant. As shown in Fig. 4(d), both variants remain stable and consistently outperform the clipping-based baseline, while the adaptive version achieves better asymptotic performance. Moreover, the learned $\lambda$ stays close to zero in the early stage and increases only later as the policy gradually moves farther from the behavior distribution, indicating that the regularization is activated mainly when stronger trust-region control becomes necessary rather than being overly aggressive from the start. This behavior is also consistent with our implementation design: for LLM reasoning we use adaptive dual updates, while for robotic control we use a fixed $\lambda$, and both settings are stable in practice. We will clarify this point in the revision to make the practical role of $\lambda$ more explicit.
>
> > Q1: Could the authors discuss scenarios where standard PPO may show superior performance or more favorable characteristics compared to R$^2$VPO?
>
> Thank you for raising this important question.
>
> First, in our continuous-control experiments, we directly compare against PPO, and we do **not** observe a regime where PPO systematically outperforms R$^2$VPO on the main benchmarks. The largest gains of R$^2$VPO appear exactly in the settings that motivate the method: sparse-reward exploration, dynamic environments, and regimes where clipping can suppress informative but high-divergence updates, such as Ball-in-Cup, CheetahRun, WalkerRun, and CartpoleSwingupSparse.
>
> That said, we agree PPO can still be attractive in regimes where these conditions are weak. In particular, when training is strictly on-policy with fresh data, rewards are relatively dense, the ratio distribution remains mild, and clipping rarely truncates useful updates, the additional benefit of variance-aware regularization may be limited. In such cases, PPO can remain competitive because of its simplicity, maturity, and ease of deployment. This interpretation is also consistent with our appendix, where we state that R$^2$VPO outperforms PPO on the **majority** of control tasks, with especially clear advantages on the harder sparse/dynamic ones, implying that the gap is smaller on easier dense-reward tasks.
>
> A similar pattern also appears in our LLM results. We compared our method against clipping-based PPO-family surrogates such as GRPO/GRPO-CH. The gains are most pronounced on fast-thinking and smaller models, where clipping is more likely to suppress rare successful exploration.
>
> We will add this discussion in the revision. More broadly, our position is that R$^2$VPO is **not** intended as a universal replacement that must dominate PPO in every regime. Rather, it is a principled alternative whose advantage becomes most evident when hard clipping itself becomes the optimization bottleneck—especially under sparse rewards, long-horizon reasoning, or stale/off-policy data reuse.

---

> > ### Author Rebuttal · Reviewer_t29D · 2026-04-04
> >
> > Thank you for the thorough and well-structured response. The clarification regarding the dual variable λ is helpful — it is reassuring to know that performance remains robust across both fixed and adaptive variants, and I appreciate the authors reminding me of this point.
> >
> > I also found the discussion on PPO's competitive regimes informative. As the authors clarified, my original question was largely out of curiosity, and I am satisfied with the explanation that the performance gap is most pronounced precisely when PPO's clipping becomes the optimization bottleneck — particularly in sparse-reward control tasks and LLM training settings. The reminder that R²VPO outperforms PPO on the majority of control tasks further solidifies my assessment.
> >
> > In light of the authors' responses, I will update my score to Accept in the final evaluation.

---

> > > ### Author Response · Authors · 2026-04-04
> > >
> > > Dear reviewer,
> > >
> > > Thank you very much for your thoughtful follow-up and for your positive acknowledgement of our rebuttal. We are very pleased to know that your concerns have been addressed, and we sincerely appreciate the time, effort, and constructive feedback you have devoted to reviewing our work.
> > >
> > > We will incorporate the discussed clarifications, related-work positioning, and additional experimental results into the final version of the paper. We are also very grateful for your indication that you plan to update your score to Accept in the final evaluation, and we truly appreciate your support.
> > >
> > > Best wishes,
> > >
> > > The authors

---

### Decision · Program_Chairs · 2026-04-30

**Decision:**

Accept (spotlight)

**Comment:**

The committee is pleased to recommend the acceptance of this paper. The paper provides a principled alternative to heuristic policy clipping with strong empirical validation in both robotics and LLM post-training. It proposes Ratio-Variance Regularized Policy Optimization ($R^2VPO$), a method that replaces standard PPO clipping with a variance-based penalty derived from the local geometry of trust regions. The reviewers consensus is that the approach is technically sound and addresses a known limitation of PPO—the suppression of rare, high-value signals in sparse-reward environments. The empirical evaluation is extensive, spanning diverse domains including Large Language Model reasoning and robotic control.

The rebuttal significantly strengthened the submission by providing additional evidence of stability in high-dimensional tasks. While the method lacks global monotonic improvement guarantees and introduces additional dual-variable hyperparameters, its performance gains in off-policy and sparse-reward settings are clear. In your final version, ensure the discussion of local versus global approximations is included, and integrate the rebuttal results regarding LLM compute efficiency into the main text to provide a complete picture for practitioners.